# In-Context Learning and Occam's Razor

**Eric Elmoznino** [* 1 2]   **Tom Marty** [* 1 2]   **Tejas Kasetty** [1 2]   **Leo Gagnon** [1 2]   **Sarthak Mittal** [1 2]   **Mahan Fathi** [3]
**Dhanya Sridhar** [* 1 2]   **Guillaume Lajoie** [* 1 2]

## Abstract

A central goal of machine learning is generalization. While the No Free Lunch Theorem states that we cannot obtain theoretical guarantees for generalization without further assumptions, in practice we observe that *simple* models which explain the training data generalize best—a principle called *Occam's razor*. Despite the need for simple models, most current approaches in machine learning only minimize the training error, and at best indirectly promote simplicity through regularization or architecture design. Here, we draw a connection between Occam's razor and in-context learning—an emergent ability of certain sequence models like Transformers to learn at inference time from past observations in a sequence. In particular, we show that the next-token prediction loss used to train in-context learners is directly equivalent to a data compression technique called prequential coding, and that minimizing this loss amounts to jointly minimizing both the training error *and* the complexity of the model that was implicitly learned from context. Our theory and the empirical experiments we use to support it not only provide a normative account of in-context learning, but also elucidate the shortcomings of current in-context learning methods, suggesting ways in which they can be improved. We make our code available at `https://github.com/3rdCore/PrequentialCode`.

## 1. Introduction

The goal of machine learning (ML) is to learn models that generalize to unseen data. Longstanding theory shows that minimizing training error alone can lead to overfitting and poor generalization (Bishop & Nasrabadi, 2006). To enable better generalization, ML follows the principle of *Occam's razor*—the best explanation is the simplest one that explains the observations (Rathmanner & Hutter, 2011; Sunehag & Hutter, 2014; Hutter, 2010). The intuition is that simple rules that explain the data cannot simply memorize observations, and must instead capture more general patterns. Consequently, learning algorithms usually trade off low training error and low model complexity with *ad hoc* approaches (e.g., via regularization and inductive biases), motivating the need for notions of complexity that can be tractably minimized directly.

Although there exist mathematical notions of model complexity such as VC dimension (Vapnik et al., 1998) or Kolmogorov complexity, these quantities cannot be directly minimized, or even tractably computed for the latter. In practice, we instead learn predictors that minimize training error as well as *proxies* of the model's complexity, such as the $L_1$ norm of the parameters, or rely on inductive biases for low-complexity solutions that are implicit in the model class and learning algorithm. Defying this trend, however, pretrained large sequence models (such as large language models—LLMs) have a surprising ability to rapidly learn and generalize from small amounts of data presented in their context (or *prompt*) without any parameter updates (Radford et al., 2019). This form of "model fitting at inference" falls under the category of *in-context learning* (ICL) which generally refers to the ability of models to learn new tasks from context. Although ICL refers to a wide range of phenomena in the literature (Lampinen et al., 2024) such as an LLM learning a task by following instructions, here, we focus on ICL as the process where a pretrained sequence model infers a statistical model from a training dataset passed in-context, which is also called *memory-based meta-learning* (e.g., Xie et al., 2022; Chan et al., 2022).

The main contribution of this paper is to provide theoretical arguments linking ICL to Occam's razor and a preference for simple models (Figure 1). Briefly, our theory frames ICL as a meta-learning algorithm whose next-token prediction objective is directly equivalent to a powerful compression method called prequential coding (Blier & Ollivier, 2018). Given the relationship between compression and Occam's razor, we show that the meta-objective in ICL is to find a

---

[*]Equal contribution  [1]Mila – Quebec AI Institute [2]Université de Montréal [3]NVIDIA. Correspondence to: Eric Elmoznino <eric.elmoznino@mila.quebec>, Guillaume Lajoie <guillaume.lajoie@mila.quebec>.

*Proceedings of the 42$^{nd}$ International Conference on Machine Learning*, Vancouver, Canada. PMLR 267, 2025. Copyright 2025 by the author(s).

**a.** Learning pipeline  **b.** Prequential Code Length  **c.** Learned prediction model

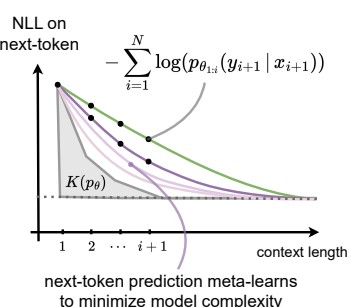

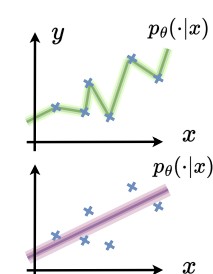

*Figure 1.* **In-context learning fits simple models to training data. a.** Standard train-risk minimization fits predictors that perform well on the training data, while a meta-learner trained on next-token prediction (in-context learning) learns to infer predictors that generalize to unseen (next-token) data, mitigating overfitting. **b.** Prequential coding: a method for estimating $K(D)$ using a model $p_\theta$'s learning algorithm $T$. As the learner $T$ sees more data, it outputs models $p_{\theta_i} = T(D_{1:i})$ that assign a higher likelihood to new observations $d_{i+1}$, and can thus better compress them. The prequential code length $L_{preq}(D;T)$ for describing the data in this way is given by the area under the curve, and the model's complexity is related to the learner's speed of adaptation. **c.** Illustration of the solutions fit by each learner. Train-risk minimization produces models that overfit the training data and generalize poorly, whereas in-context learning infers simpler solutions.

learner capable of jointly minimizing both training error *and* model complexity across a diverse range of tasks. Our theory, along with the empirical experiments that we use to support it, explain why ICL has proven so effective in meta-learning settings, and also explain the shortcomings of current ICL methods. Namely, we find that current methods produce learning algorithms which are susceptible to underfitting and can fail to generalize to novel tasks, suggesting principled avenues for future research.

## 2. In-Context Learning and Occam's Razor

In this section, we introduce a meta-learning objective that targets simple models, and then show that it is equivalent to the next-token prediction objective underlying ICL. We reach this result via four key steps:

1. We begin by formalizing both training error and model simplicity through the lens of Kolmogorov complexity, which deals with optimal data compression.

2. We then show how learning algorithms can be used to compress data through a technique called prequential coding (Blier & Ollivier, 2018), whose resulting "prequential code length" can be minimized to improve compression.

3. We then introduce the idea of finding a learning algorithm that minimizes prequential code length by formalizing a meta-learning problem that appears difficult to optimize.

4. Finally, we show that the next-token prediction objective underlying ICL *already* solves this meta-learning

problem in an efficient and scalable way.

### 2.1. Kolmogorov Complexity and Data Compression

Kolmogorov complexity (Kolmogorov, 1965; Li & Vitányi, 2008) is a notion of information quantity. Intuitively, the Kolmogorov complexity $K(x)$ of an object $x$ is the length of the shortest program (in some programming language) that outputs $x$. A related notion is the conditional Kolmogorov complexity $K(x|y)$ of the object $x$ given another object $y$, which is the length of the shortest program that takes $y$ as input and outputs $x$. While quite abstract, this notion of complexity has deep ties to *compression*, making it intuitive as a measure of information quantity. The smaller and more "structured" an object is—regularity, patterns, rules, etc.—the more easily it can be described by a short program, correspondingly having lower Kolmogorov complexity. Although Kolmogorov complexity is very general—objects $x, y$ can be datasets, programs, models—it is intractable to compute. However, it can often be tractably estimated or bounded.

A quantity relevant to ML is the Kolmogorov complexity of a dataset $K(D)$, where $D = (d_1, ..., d_n)$ with $d_i \sim p$. According to (Grünwald, 2007), if the dataset is sufficiently large, it can be optimally compressed by first encoding the data-generating process $p$ and then encoding the data under this distribution. It is well known that optimally encoding data under a distribution takes only $-\log_2 p(D)$ bits (e.g., using an arithmetic coding scheme, Witten et al., 1987), as in the case of Shannon information (Shannon, 2001). We therefore have that:

$$K(D) = K(p) + K(D|p) = K(p) - \log_2 p(D), \quad (1)$$

where $K(p)$ refers to the complexity of the distribution (i.e., the length of the shortest program that outputs function $p : \mathcal{D} \to \mathbb{R}^+$). This term is intractable to compute as it requires an enumeration over all programs that output $p$, but $K(D|p) = -\log_2 p(D)$ is easily computed: it is simply the negative log-likelihood of the data under $p$, a commonly used objective function in ML. It follows that simple models which achieve lower training error better compress data. We provide further background on Kolmogorov complexity in Appendix A.

As we are interested in model optimization, we henceforth consider parameterized models $p_\theta$ with parameters $\theta$. We denote a learning algorithm by a function $T : \mathcal{P}(\mathcal{D}) \to \Theta$ (where $\mathcal{P}$ denotes the power-set), which maps a dataset $D$ to a model $p_{T(D)}$. Maximum likelihood training on *iid* data, which is the norm in ML, is a learning algorithm $T^{ml}$ which minimizes:

$$T^{ml}(D) = \arg\min_{\theta'} - \sum_{d \in D} \log_2 p_{\theta'}(d). \qquad (2)$$

However, Occam's razor says that we also need simple models that *best compress the data*–mathematically, this is because the "true" model generating the data in Equation (1) *optimally compresses the data*. Thus, we consider the learning algorithm $T^{oc}$, which defines "simple" via complexity:

$$T^{oc}(D) = \arg\min_{\theta'} \left[ K(p_{\theta'}) - \sum_{d \in D} \log_2 p_{\theta'}(d) \right]. \qquad (3)$$

In reality, the Occam's razor learner $T^{oc}$ is intractable since $K(p_{\theta'})$ cannot be computed. In practice, maximum log-likelihood training $T^{ml}$ is often enhanced with regularizers (e.g., $L_2$-norm of parameters) and inductive biases (e.g., restricting the model class) to implicitly favor low-complexity models that combat overfitting and improve generalization. For instance, deep neural networks (DNNs) trained through stochastic gradient descent (SGD) tend to be biased towards simple solutions (Blier & Ollivier, 2018; Goldblum et al., 2024; Mingard et al., 2025). However, most existing regularizers at most amount to *indirect* methods that roughly penalize model complexity $K(p_\theta)$ along with training error; learning algorithms (which we will often call "learners" for brevity) rarely *directly* attempt to minimize compression length of $D$, as $T^{oc}$ would. In what follows, we introduce learners $T_\phi$ that have learnable parameters $\phi$, estimated via meta-optimization, to approximate the ideal learner $T^{oc}$.

## 2.2. Prequential Coding

While a learner $T$ that adheres to Occam's razor and solves Equation (3) by optimally compressing $D$ would improve generalization, it is difficult to design one in practice. Even if $K(p_\theta)$ could be computed efficiently, there is the further challenge of minimizing it. Instead, we will first describe an algorithm for efficiently compressing $D$ that does not require estimating $K(p_\theta)$, and then we will consider how to optimize this compressor in the next section.

While $K(p_\theta)$ is difficult to measure directly, it turns out that we can compress $D$ without it using an algorithm called *prequential coding* (illustrated in Figure 1) that leverages the learner $T$ which produced $p_\theta$ (i.e., $p_\theta = T(D)$). Consider an ordered dataset $D = \{d_1, ..., d_N\}$, and denote $D_{1:i} = \{d_1, ..., d_i\}$. Prequential coding uses the learner $T$ to train models on increasing amounts of data. First, we train a model on just the first data point to get $p_{\theta_1} = T(d_1)$. Because the model is trained on a single datapoint, it will not be very accurate; however, it should be better than a random model that has seen no data at all. We can then use this model $p_{\theta_1}$ to compress the next (unseen) datapoint $d_2$, which takes $-\log_2 p_{\theta_1}(d_2)$ bits. At this point, we can train a new model $p_{\theta_2} = T(D_{1:2})$. Having seen more data, this model should assign a higher likelihood to a new datapoint $d_3$, which we can compress using $-\log_2 p_{\theta_2}(d_3)$ bits. This process repeats until the entire dataset has been covered. At this point, the model $p_\theta$ can be obtained simply by applying the learning algorithm to the complete dataset $p_\theta = T(D)$.

The total number of bits that it takes to compress $D$ using prequential coding is the sum of how many bits it takes to compress each datapoint using a model that was trained on all previous ones. Visually, it is the area under the *prequential coding curve* shown in Figure 1b. The length of this program is called the *prequential code length* $L_{preq}(D; T)$ (Blier & Ollivier, 2018):

$$L_{preq}(D; T) = \sum_{i=0}^{N-1} -\log_2 p_{\theta_i}(d_{i+1}) \geq K(D). \qquad (4)$$

$L_{preq}(D; T)$ is an upper-bound on $K(D)$ since prequential coding is one way to compress the data. In Section 2.3 we will minimize this quantity with respect to the learner $T$, and thus minimize the description length of the data.

Prequential coding relates Kolmogorov complexity to intuitions about generalization in ML: the simpler a model is, the quicker it generalizes from limited amounts of training data. Although the relationship in Equation (4) offers a promising way forward to operationalize Occam's razor, there is a problem. The prequential code length given by Equation (4) *conditions* on the choice of a learner $T$. However, prequential coding also requires us to encode the learning algorithm itself. When we take the description length of $T$ into account, the quantity $L_{preq}(D; T) + K(T)$ upper bounds $K(D)$ (see Appendix B). As we describe below, we will optimize for learners $T_\phi$ that minimize $L_{preq}(D; T_\phi)$, and will articulate how $T_\phi$ has low complexity given minimization over multiple datasets.

## 2.3. Minimizing Prequential Code Length

Consider a parameterized learner $T_\phi$ that minimizes the prequential code length $L_{preq}(D; T_\phi)$ of a dataset $D$. This objective aims to optimally compress $D$ like the idealized learner $T^{oc}$ does, but only when $K(T_\phi)$ is low. This second criteria is violated if $T_\phi$ overfits to a single dataset $D$. To forbid $T_\phi$ from memorizing a single dataset, we consider a meta-dataset $\mathscr{D} = \{D^1, ..., D^M\}$ coming from $M$ different tasks and meta-learn $T_\phi$ to minimize prequential code length on average across the meta-dataset $\mathscr{D}$. This allows us to write the following objective for the learner $T_\phi$:

$$\mathcal{L}(\mathscr{D}; \phi) = \sum_{i=1}^{M} L_{preq}(D^i; T_\phi) \geq \sum_{i=1}^{M} K(D^i | T_\phi). \quad (5)$$

By minimizing $\mathcal{L}(\mathscr{D}; \phi) = \sum_{i=1}^{M} L_{preq}(D^i; T_\phi)$, we thus end up with a learner $T_{\phi^*}$ that minimizes compression length in expectation over datasets, and after meta-training compresses a new dataset of interest $D$ using $L_{preq}(D; T_{\phi^*}) \geq K(D)$ bits. The learner $T_{\phi^*}$ will succeed in compressing $D$ when it *generalizes* to that novel dataset.

The learner $T_{\phi^*}$ and the Occam's razor learner $T^{oc}$ ($= \arg\min_{\theta'} [K(p_{\theta'}) - \log_2 p_{\theta'}(D)]$) are not exactly identical: $T^{oc}$ compresses the data by directly minimizing model complexity and training error, whereas $T_{\phi^*}$ compresses the data by minimizing its prequential code length. Despite these differences in compression strategy, however, the two learners are deeply related through the minimum description length principle (Grünwald, 2007) because they both attempt to optimally compress their training data. In particular, if $T_{\phi^*}$ significantly compresses the data through prequential coding, the minimum description length principle states that the model $p_\theta = T_{\phi^*}(D)$ which it fits is likely to generalize because optimal compression is achieved by the true generative process (Equation (1)). In this way, the learner $T_{\phi^*}$ implicitly fits simple models with low training error, minimizing $K(p_\theta) - \log_2 p_\theta(D)$. In Section 3, we will empirically show this to be the case across diverse tasks, where $p_\theta = T_{\phi^*}(D)$ will generalize far better than the typical ML approach of minimizing training error alone.

## 2.4. ICL Minimizes Prequential Code Length

In practice, solving the meta-learning problem in Equation (5) involves several constraints:

1. The performance of $T_\phi(\cdot)$ must be evaluated w.r.t. a dataset's prequential code length.

2. $T_\phi(\cdot)$ must be fast to evaluate because it is iteratively called on multiple datasets.

3. To meta-optimize $\phi$, it must be easy to take gradients of $L_{preq}(\cdot; T_\phi)$ w.r.t. $\phi$.

4. $\phi$ must parameterize an expressive class of learning algorithms, capable of minimizing prequential code length on a broad distribution of tasks and generalizing to unseen ones.

While this may appear daunting, it turns out that these desiderata are readily addressed by ICL in probabilistic sequence models. Such models are trained to predict the distribution over the next element in a sequence given its past context: $F(d_t | D_{1:t-1})$. Crucially, the sequence model $F$ is *both* the learner $T_\phi$ and the inner model $p_\theta$. Indeed, $\phi$ corresponds to the parameters of the sequence model $F$ (e.g. weights in a Transformer), and $\theta = T_\phi(D_{1:t-1})$ is encoded by the activations of hidden units in the model when presented with the context $D_{1:t-1}$. Thus, the predicted distribution over the next token is given by: $F(d_t | D_{1:t-1}) = p_{T_\phi(D_{1:t-1})}(d_t)$. The model is trained to minimize the cumulative next-token prediction error: $\mathcal{L}(D; \phi) = \sum_{t=1}^{N} -\log p_{T_\phi(D_{1:t-1})}(d_t)$, which corresponds exactly to the prequential code length in Equation (4).

The dual nature of the sequence model as both the learner and the learned model offers a natural solution to the constraints above, enabling fast and differentiable evaluation of $T_\phi(\cdot)$ (2 & 3 above) with respect to cumulative next-token prediction loss (1 above). Moreover, modern sequence models can parameterize a rich class of learning algorithms, which is crucial to minimizing Equation (5) (4 above). Notably, architectures such as Transformers are known to have components which make them especially good meta-learners, such as multi-head attention (Olsson et al., 2022). It is thus no surprise that sequence models are leveraged in settings outside of the language domain (Von Oswald et al., 2023; Bauer et al., 2023; Kirsch et al., 2022), making them general-purpose meta-learners.

This predictive formulation can be used to model data which contains sequential correlations, such as language, but it is more general. Indeed, consider an *iid* dataset $D = \{(x_1, y_1), ..., (x_T, y_T)\}$ and the supervised task of learning a function $y = f(x)$. In this setting, a data point is given by the pair $d_t = (x_t, y_t)$, and straightforward tokenization schemes can be used to append a novel query $x^*$ to the context $D$ such that the predicted output $\hat{y}^*$ is given by the next token in the sequence. This ICL setup is well-suited for regression-type tasks (see e.g. (see e.g., Von Oswald et al., 2023; Oswald et al., 2023)) but can be used for most supervised tasks. ICL thus turns the training of a sequence model into a meta-optimization problem over datasets—an approach also called *memory-based* meta-learning (Hochreiter et al., 2001; Santoro et al., 2016; Ortega et al., 2019).

Prequential code length is well-defined over arbitrary sequences, so our theory relating ICL to Occam's razor applies to both *iid* and nonstationary data, as explored in Section 3.

**Summary.** We showed that sequence models trained on cumulative next-token prediction losses explicitly optimize a meta-learning objective for compression, thus jointly minimizing training error and model complexity. This provides a normative account of ICL in terms of Occam's razor, and explains recent experimental findings showing that LLMs are good universal compressors (Deletang et al., 2024).

## 3. Experiments

Our experiments are designed to illustrate the benefits of ICL in terms of fitting simple models that generalize. In Section 3.1, we compare ICL's standard next-token prediction objective to an alternative that minimizes training error alone, rather than prequential code length. Section 3.2 then compares ICL to standard gradient-based learners that minimize training error, such as SGD. Section C.2 shows the impact of regularization on gradient-based learners from a compression perspective. In Section 3.3, we explore the impact of learner $T_\phi$'s architecture on prequential code length minimization. Section 3.4 explores the ability of $T_\phi$ to generalize to novel tasks. Experimental details not described in the main paper (e.g., precise architectures, hyperparameters, etc.) can be found in Appendix C.

**Tasks.** In line with similar work studying ICL in a controlled setting (Mahankali et al., 2024; Garg et al., 2022; Akyürek et al., 2023), we use synthetically-generated tasks. Each task consists of a supervised learning dataset $D^i = \{(x_1, y_1), ..., (x_k, y_k)\}$, where the labels are a (potentially stochastic) function of the input $y_j = f^i(x_j, \epsilon_j)$. ICL learners $T_\phi$ are trained on a meta-dataset $\mathscr{D} = \{D^1, ..., D^N\}$, where each $D^i$ is associated with a different ground-truth data-generating function $f^i$. We primarily study three meta-datasets: **(1) Linear regression** problems where $x \in \mathbb{R}^3$ and $y \in \mathbb{R}$. The ground-truth functions $f^i$ are noisy linear mappings $y_j = W^i x_j + b^i + \epsilon_j$, where each $\{W^i, b^i\}$ is sampled from a standard Normal distribution and $\epsilon_j$ is Gaussian noise with $\sigma^2 = 0.04$. **(2) Sinusoidal regression** problems where $x_j \in \mathbb{R}$ and functions $f^i$ are linear combinations $y_j = \sum_{l=1}^{L} \alpha^{i,l} \sin(\omega^l x_j)$. We use $L = 3$ with frequencies $\omega^l \sim U(0, 5)$ that are shared across tasks, varying only the amplitudes $\alpha_{i,l} \sim \mathcal{N}(0, 1)$. **(3) Mastermind**: a multi-label classification problem inspired by the code-breaking game *Mastermind*. Each $f^i$ is associated with an underlying discrete code (a fixed-size sequence of digits) that needs to be inferred from random guesses that return partial information. The inputs $x_j$ are random guesses for the code, and $y_j = f^i(x_j)$ is a tuple of two class labels where the first specifies the number of digits in $x_j$ that are correct in terms of both position and value, and the second label specifies the

number of digits that are correct in value but not necessarily position. We use randomly sampled codes of length 8 with digits varying from 1..6.

The tasks above produce *iid* datapoints so that we could make fair comparisons to learners that minimize training error under *iid* assumptions (e.g., SGD). However in Section 3.3 we will compare prequential ICL learners with different architectures, and we consider another, *non-iid* task, **(4) HMM:** next token prediction on synthetically-generated nonstationary data from Hidden Markov Models (HMMs) that were designed to mimic the statistical properties of natural language in a simplified and controlled setting. The models are evaluated on unseen HMMs with novel transition and emission matrices. See Appendix C for details.

### 3.1. Comparisons to ICL With a Train-Risk Objective

We have argued that standard ICL can be seen as a meta-learning method who's meta-objective is to minimize training error and model complexity through cumulative next-token prediction (prequential code length). However, this is not the only meta-objective that one could design for ICL. In particular, we can design an alternative meta-objective that minimizes *only* training error simply by training $T_\phi$ to predict *past* datapoints in the context rather than future unseen ones. In both cases, the learner $T_\phi$ is some function that takes a context (i.e., a partial dataset) as input, and outputs a model $p_\theta$ capable of making predictions for arbitrary datapoints. For supervised learning, this can be represented as $\hat{y}_q = T_\phi((x, y)_{1:j}, x_q)$ where $(x, y)_{1:j}$ corresponds to an observed context, $x_q$ is the queried input, and the model $p_\theta$ is implicitly encoded in $T_\phi$'s weights and latent activations given the context. In standard ICL (which we will refer to as *prequential ICL*), the query $x_q$ is a novel input that does not appear in the context. In the alternative form of ICL (which we will call *train-risk ICL*), the query $x_q$ is a randomly-selected input that appeared previously in the context $x_{1:j}$. Note the similarities of train-risk ICL to standard objectives of learners that minimize training error: it processes some fixed-sized training set (here a context) and attempts to minimize the empirical risk on a subset of that very same data (here a single query that appeared in the context). While nobody uses train-risk ICL in practice, it serves as an ideal control to illustrate our theory of ICL and the generalization benefits of minimizing prequential code length as opposed to only training error. One can use an identical architecture for $T_\phi$ in both cases and train using precisely the same methodology and loss function; the only difference is which query the loss function is evaluated on.

In our experiments, we parameterize $T_\phi$ using a Transformer. For the train-risk case, a standard Transformer could simply attend to the context position that matches $x_q$ and retrieve the corresponding label. To prevent this trivial solution,

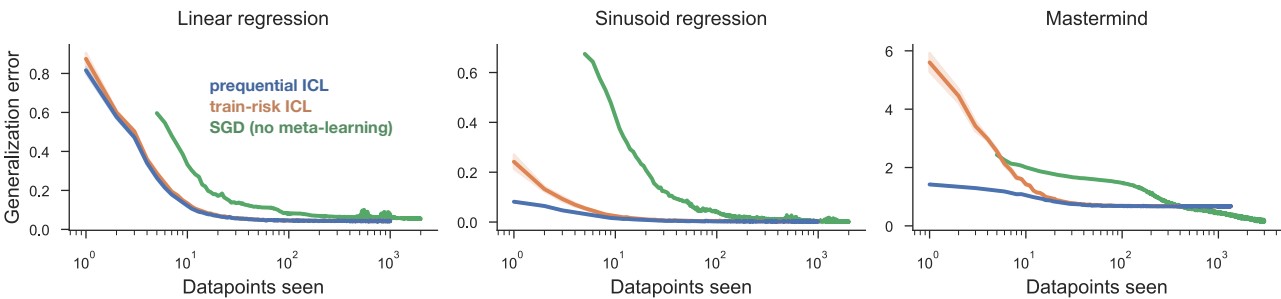

*Figure 2.* **Experimental results comparing different learners.** Figures show average prequential coding curves for a meta-dataset, which is the mean prediction error on unseen data (generalization error, y-axis) given observed contexts of increasing length (datapoints seen, x-axis). The area underneath these curves corresponds to prequential code length. ICL from next-token prediction objectives (prequential ICL, blue) yields lower prequential code lengths than ICL from past-token prediction objectives (train-risk ICL, orange), with greater effects in low-data regimes. An SGD-based learner (green) fits more complex models than prequential ICL and performs poorly in low-data regimes, but can generalize better in large-data regimes on a difficult Mastermind task due to underfitting in ICL. Error is measured using MSE for linear and sinusoid regression and cross-entropy for Mastermind. Error bars show standard error across seeds (5 for ICL, 15 for SGD).

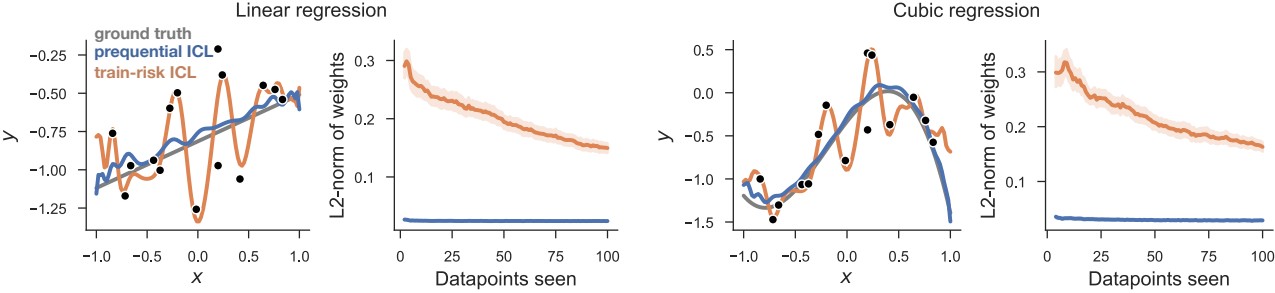

*Figure 3.* **Models inferred by different learners.** Visualization of inferred models for noisy linear and cubic regression tasks (leftmost and middle right). Observed data is shown in black points, ground-truth in gray, prequential ICL in blue, and train-risk ICL in orange. Train-risk ICL consistently overfits, while prequential-ICL infers simpler functions closer to the ground truth. Middle left and rightmost plots show the $L2$-norm of inferred coefficients for polynomial orders greater than the ground-truth data-generating orders, plotted against context length. Train-ICL fits complex models using high-degree components to overfit the data.

we instead use a bottlenecked architecture for $T_\phi$ described in Mittal et al. (2025). In this architecture, a Transformer first summarizes the context into a low-dimensional vector $z = \texttt{Transformer}_\phi((x, y)_{1:j})$, and a separate prediction head—here a multi-layer perceptron (MLP)—subsequently outputs a prediction for the query $\hat{y}_q = \texttt{MLP}_\phi(x_q, z)$. For fair comparison, we use the same bottleneck architecture for train-risk ICL and prequential ICL in all experiments, unless otherwise stated. Figure 2 shows our comparisons between prequential ICL to train-risk ICL, where we plot the prequential coding curves for each ICL method after loss convergence on a meta-dataset. The curves are constructed at inference time by evaluating the average generalization error (i.e., unseen next-token prediction loss) on *unseen* tasks from the meta-dataset, for varying context lengths.

**Findings.** Two findings follow directly from our theory. The first is that for large context lengths, generalization error is identical for both prequential ICL and train-risk ICL.

This is because with significant data, optimal compression is dominated by training error (Equation (1)), making overfitting less likely to occur. The benefits of simple models are instead expected to be most prominent in *low-data* regimes where generalization is difficult, and this is precisely what we observe. Across all tasks, prequential ICL consistently outperforms train-risk ICL in terms of generalization for short context lengths, and this performance gap extends further the more difficult the task (e.g., it is small for linear regression, and larger for sinusoid regression and mastermind). We confirm that the performance gap widens with increasing task difficulty by fixing the function class and increasing the dimensionality of the inputs $x$ in Appendix D, which is expected given that harder tasks require more data for generalization. In Appendix F we repeat these analyses on a modified version of the HMM task that is more amenable to train-risk minimization and find similar trends as above for *non-iid* data. Lastly, in Appendix G, we com-

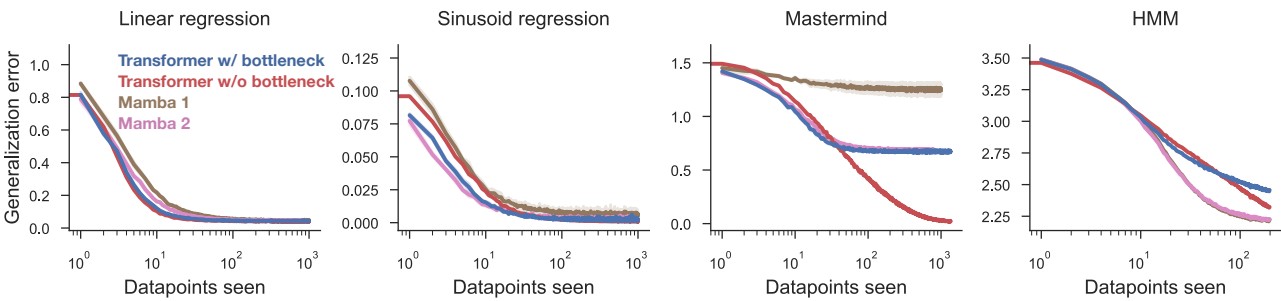

*Figure 4.* **Sequence model architecture impacts ICL's ability to minimize prequential code length.** Figures show average prequential coding curves for a meta-dataset, which is the mean prediction error on unseen data (generalization error, y-axis) given observed contexts of increasing length (datapoints seen, x-axis). The area underneath these curves corresponds to prequential code length. Error is measured using MSE for linear and sinusoid regression and cross-entropy for Mastermind and HMM. Error bars show standard error across 5 seeds.

pare the generalization performance of an in-context learner trained on the full sequence of tokens versus only the latter half of the sequence. Consistent with our theory, the lack of low-data signal in the latter case results in worse prequential code length and less pressure toward simple models.

**Visualization of learned models.** To better understand the solutions learned by prequential and train-risk ICL, we visualize the inferred models for noisy linear and cubic regression tasks in Figure 3. We also replace the prediction head of our bottlenecked models with a parameter-free polynomial function, so that the bottleneck represents a vector of inferred polynomial components. Figure 3 shows that for a small context length (15) train-risk ICL overfits the data by fitting complex models with high polynomial order, whereas prequential ICL fits simple functions that are more robust to noise. Further experimental details are provided in Section C.3.

### 3.2. Comparisons to Gradient-Based Learners

We next consider whether there are empirical advantages of meta-learning a learner $T_\phi$ to minimize prequential code length through ICL, compared to using standard out-of-the-box learning algorithms. In particular, we know that traditional SGD-based learners can optimize DNN models that generalize well across a wide range of tasks, despite only explicitly minimizing training error. We consider a standard SGD-based learner that fits a randomly-initialized MLP to the training set until validation loss converges. We repeatedly sample a dataset from our meta-dataset, truncate it to a specified number of observed datapoints, apply the SGD-based learner to the truncated dataset, and evaluate the resulting model's generalization error on new datapoints.

**Findings.** Figure 2 compares this SGD-based learner to prequential (and train-risk) ICL learners. Across all tasks, the models obtained through ICL generalize better in low-data regimes, aligning with our theory that ICL minimizes model complexity. With enough training data, however,

models obtained through the SGD-based learner generalize just as well. In fact, on the Mastermind task, SGD performs *better* in large-data regimes. This result demonstrates that even though the next-token prediction objective in ICL is well-motivated from the theoretical perspective of compression, the degree to which that objective can successfully be minimized strongly depends on the architecture of $T_\phi$ and the methods used to train it. For instance, when $T_\phi$ is a Transformer, the expressivity of the model it implicitly fits to the context scales with the number of activations in the network ($N$), whereas the expressivity of a DNN trained through SGD scales with the number of weights ($N^2$). Furthermore, the amount of compute time that $T_\phi$ uses to fit the context amounts to one forward pass of a network, whereas the compute time that goes into fitting a dataset using SGD can be arbitrarily large.

**Regularized SGD.** We also considered the effects of different regularization techniques that can be applied to SGD to indirectly influence the complexity of the resulting model (e.g., $L_2$), shown in Appendix E. As expected, regularization influences prequential code length by trading off model complexity and training error to varying degrees. However, all SGD regularization methods are still outperformed by prequential ICL in low-data regimes, due to prequential ICL's direct compression objective that optimally balances model complexity and training error.

### 3.3. Influence of the ICL Architecture

The previous section argued that the structure of $T_\phi$ can influence its ability to minimize prequential code length. In this section, we further illustrate this point by considering a wider breadth of neural architectures for $T_\phi$. Since state-space models (SSMs) have recently been shown to exhibit ICL (Lu et al., 2024), we test Mamba 1 (Gu & Dao, 2024) and Mamba 2 (Dao & Gu, 2024). We also test a standard causal Transformer in addition to the bottlenecked Transformer from previous sections. We refer to Appendix C for

additional information about the specificity of each architecture.

**Findings.** Prequential code length comparisons in Figure 4 show that the architecture for $T_\phi$ indeed plays a substantial role, which interacts substantially with the meta-dataset. For instance, only the Transformer without bottleneck does well on Mastermind, whereas on the HMM task it is outperformed by SSMs. Analyzing these results in depth is out of scope for this work; we only intend to show that having a next-token prediction objective alone does not guarantee that prequential code length can successfully be minimized in practice through ICL.

### 3.4. Large Pretrained Models

A core element of our theory of ICL is that $T_\phi$ is trained to minimize average prequential code length on a meta-dataset. There is no guarantee, however, that prequential code length will be small on a novel dataset at inference time: this depends on the generalization abilities of the learner $T_\phi$. In this section, we look at the task-generalization abilities of a large pretrained LLM (GPT-4 Achiam et al., 2023) on the Mastermind task. We do this by prompting the LLM with a description of the task and a number of in-context examples, then obtaining the logits and prediction error for a novel example.

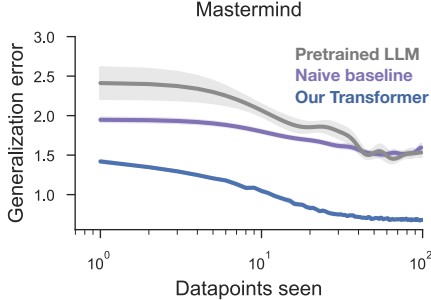

*Figure 5.* **Large pretrained sequence models can fail to minimize prequential code length on novel tasks.** GPT-4 (gray) performs far worse than small ICL models trained on a distribution of Mastermind tasks (blue) and a naive baseline that predicts the marginal class distribution over the context (purple). Error is measured using cross-entropy. Error bars show standard error across 5 seeds.

**Findings.** In Figure 5, we find that despite its massive pretraining across a breadth of tasks, the LLM is unable to meaningfully minimize prequential code length on Mastermind. Not only is its prequential code length substantially higher than for a much smaller model trained on a distribution of Mastermind tasks, but it is also higher than for a naive baseline that just predicts the empirical marginal distribution over class labels in the context. These results demonstrate that even when the size of the model and meta-

dataset used to train $T_\phi$ are scaled significantly, current methods for ICL can still struggle to minimize prequential code length on a novel task.

## 4. Related Work

**Sequence modeling and compression.** The idea that probabilistic models can be used to efficiently compress data is a topic widely studied in machine learning across different modalities and settings (Ollivier, 2015; Deletang et al., 2024; Blier & Ollivier, 2018; Veness et al., 2015), specifically in sequence modeling (Goyal et al., 2019; Valmeekam et al., 2023; Deletang et al., 2024) due to its close similarities to prequential coding (Blier & Ollivier, 2018). While Goyal et al. (2019) and Valmeekam et al. (2023) claim that learned sequence models can outperform simple compressors like JPEG or gzip, they overlook model complexity in their analysis, adhering strictly to Shannon's notion of compression. In contrast, more recent studies from Deletang et al. (2024) and Bornschein et al. (2023) opted for the Kolmogorov approach, incorporating model size to account for model complexity. Deletang et al. (2024), in particular, add nuance to the claimed advantages of foundation models due to the substantial memory allocation required to store their weights. Our theory builds on these works by relating compression and sequence modeling to the approach of meta-learning across tasks using ICL, which we show yields simple models that adhere to Occam's razor.

**ICL as Bayes-optimal prediction.** One of the dominant perspectives of ICL and related meta-learning approaches is that they yield Bayes-optimal learners (Ortega et al., 2019; Mikulik et al., 2020; Müller et al., 2022; Hollmann et al., 2023; Binz et al., 2023; Wang et al., 2023), in the sense that they learn a prior distribution over tasks during training, and then compute a posterior given data presented in-context at inference time. This posterior can then be used to make predictions with minimum Bayes' risk. Various studies have tested this in controlled settings with tractable posteriors (Xie et al., 2022; Panwar et al., 2024; Genewein et al., 2023; Mittal et al., 2023). Xie et al. (2022) assume a *concept* latent that parameterizes the generation of dependent samples through an HMM and provide formal conditions for ICL to effectively approximate the Bayes-optimal predictor on the prompt. In a supervised fashion, Akyürek et al. (2023) construct sequence of labeled examples $(x, f(x))$ and shows that under uncertainty, ICL behaves as the Bayes-optimal predictor on noisy linear regression. Additionally, they argue that with limited capacity, ICL does not necessarily match the Bayes predictor but can meta-learn other learning algorithms, such as gradient-based algorithms on linear models and closed-form ridge regressors (Panwar et al., 2024). Grau-Moya et al. (2024) induce a prior for model simplicity in ICL by generating tasks from short programs run on Uni-

versal Turning Machines. Finally, (Raventós et al., 2024) find that under a sufficiently diverse set of pretraining tasks, ICL does *not* yield Bayes-optimal predictors, but instead infers a more uniform prior. While the Bayesian perspective of ICL is very useful and complementary to the Kolmogorov one that we have proposed, we argue in Appendix H that the Kolmogorov perspective generalizes the Bayesian one and more easily accounts for diverse findings in ICL (e.g., cases where ICL does not yield Bayes-optimal predictors).

**ICL as a direct meta-learned optimizer.** Elaborating on the possibility that ICL emulates non-Bayesian learning algorithms, Von Oswald et al. (2023) show that $k$-layer linear Transformers with a specific weight parameterization can mimic $k$ steps of gradient descent for a least squares loss. Ahn et al. (2023) provide a theoretical foundation for these observations, provably showing that the optimization of the parameters of a linear Transformer under certain assumptions about the data distribution effectively implements this learning algorithm. Concurrent studies by Zhang et al. (2024) and Mahankali et al. (2024) report similar findings, albeit under slightly different assumptions regarding weight initialization or data generation processes. Beyond the scope of linear regression, Kirsch et al. (2022) explore this phenomenon on augmented natural data (MNIST, CIFAR10) and provide insightful empirical conditions for the emergence of ICL as a general-purpose learning algorithm. Other works empirically show that Transformers can learn more complex function classes in-context, such as sinusoidal regression (Von Oswald et al., 2023), decision trees (Garg et al., 2022), and RASP-programmable functions (Zhou et al., 2024). While prior works such as these attest to the powerful meta-learning capabilities of ICL, our work differs in that it identifies the precise meta-*objective* as an implementation of Occam's razor.

## 5. Discussion and Future Work

In this work, we introduced novel theoretical arguments linking ICL and the next-token prediction objective to Occam's razor. Our theory provides a normative account of the strong generalization abilities of in-context learners at inference time, especially in low-data regimes when compared to traditional optimizers. These theoretical insights were supported by a number of empirical experiments, some of which also identified shortcomings of current methods for ICL that should be addressed in future work.

One such shortcoming is that models learned through current ICL methods can underfit data presented in-context, which can hamper generalization in large-data regimes on difficult tasks. We also found that the degree of underfitting was highly dependent on the architecture used to parameterize the in-context learner (i.e., the sequence model)—a

finding corroborated by Ding et al. (2024). In light of this, we hypothesize that ICL can be improved through the design of novel sequence model architectures that explicitly target prequential code length. For example, current methods learn in-context through a single forward pass of a sequence model with fixed layer depth. In contrast, DNNs can be trained using gradient-based methods until training loss converges, which can take weeks and substantial compute. One improvement to ICL might therefore be to augment current sequence model architectures with "layers" that use built-in optimization primitives with variable compute budgets, as was done in Oswald et al. (2023). Another promising approach is to combine ICL and SGD through a "mixture of learners" that reaps their complementary benefits. ICL is sample-efficient and generalizes well in low-data regimes, while SGD-based methods that optimize the weights of a DNN excel on difficult tasks when significant training data is available. Recent work by Bornschein et al. (2024) explored a simple method for combining both learners by presenting a smaller number of *recent* tokens in-context to a sequence model for ICL, while at the same time using a large number of earlier tokens to fine-tune the weights of the sequence model using gradient methods, finding significant performance gains.

Another challenge of ICL that follows directly from our theory is that the in-context learner must generalize to novel tasks and datasets. While we found that task generalization was successful over narrow task distributions (e.g. a distribution of linear regression tasks), we also found that task generalization was more difficult in open-ended cases, in which even a large pretrained LLM was unable to learn in-context on a novel task that was easily solved by a small MLP trained using SGD. One possible path forward is to have many domain-specific in-context learners that each specialize in compressing data from a given task distribution. Another option is to learn *simple learners* that are more likely to generalize to novel tasks, which could be achieved through inductive biases, regularization, or, intriguingly, through an additional meta-layer of ICL at the task level that would minimize the Kolmogorov complexity of the learner itself (and not only the model it fits).

Finally, given that our theory generalizes to nonstationary data (where prequential coding remains a strong compression algorithm), many further questions arise. In particular, the ordering of data presented in-context can have a substantial impact on prequential code length and model complexity (Zhang et al., 2020), which can guide the design of optimal curricula for strong and data-efficient generalization.

## Acknowledgments

The authors would like to acknowledge the following researchers for valuable discussions and exchanges: Joao Sacramento, Johannes von Oswald, Jorg Bornschein, Marcus Hutter. All authors acknowledge an unrestricted gift from Google inc. for research support. EE acknowledges support from Vanier Canada Graduate Scholarship #492702. SM acknowledges the support of PhD Excellence Scholarship from UNIQUE. DS acknowledges support from NSERC Discovery Grant RGPIN-2023-04869, and a Canada-CIFAR AI Chair. GL acknowledges support from NSERC Discovery Grant RGPIN-2018-04821, the Canada Research Chair in Neural Computations and Interfacing, and a Canada-CIFAR AI Chair.

## Impact Statement

Hundreds of millions of people now use conversational agents weekly for productivity, learning, and more. These agents rely heavily on In-Context Learning (ICL) to generate high-quality, contextually appropriate responses. However, as their use expands into high-stakes settings, understanding ICL's inner workings and failure modes becomes critical to ensure the safe deployment of conversational AI. Our work aims to uncover some of the characteristics and limitations of a particular form of ICL, providing valuable insights for developing more robust ICL in future work. Such advancements are crucial for accelerating the path toward safe general artificial intelligence (AGI). The societal implications of AGI are broad and significant, and we direct readers to the field of AI safety for further discussion on these critical issues.

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

# A. Background on Kolmogorov complexity

Kolmogorov complexity was independently developed in the 1960s by Kolmogorov (1965), Solomonoff (1964), and Chaitin (1966), and defines a notion of "information quantity".

Intuitively, the Kolmogorov complexity of an object is the length of the shortest program (in some programming language) that outputs that object. Specifically, given some finite string $x$, $K(x)$ is the length $l(r)$ (in bits) of the shortest binary program $r$ that prints $x$ and halts. Let $U$ be a universal Turing machine that executes these programs. The Kolmogorov complexity of $x$ is then:

$$K(x) = \min_r \{l(r) : U(r) = x, r \in \{0,1\}^*\}, \tag{6}$$

where $\{0,1\}^*$ denotes the space of finite binary strings. A related notion is the conditional Kolmogorov complexity of a string $x$ given another string $y$, which is the length of the shortest program that takes $y$ as input and outputs $x$:

$$K(x|y) = \min_r \{l(r) : U(r(y)) = z, r \in \{0,1\}^*\}, \tag{7}$$

where $r(y)$ denotes a program taking $y$ as input. Finally, we can also define a "joint" Kolmogorov complexity $K(x, y)$, which denotes the length of the shortest program that jointly outputs both $x$ and $y$. Surprisingly, joint Kolmogorov complexity is related to conditional Kolmogorov complexity (up to an additive logarithmic term, which we will ignore) by the Symmetry of Information theorem (Li & Vitányi, 2008):

$$K(x, y) = K(y|x) + K(x) = K(x|y) + K(y). \tag{8}$$

Kolmogorov complexity has many intuitive properties that make it attractive as a measure of information quantity, and although it is less common than notions from Shannon information theory (Shannon, 2001), it is strictly more general (as we will show later below). The smaller and the more "structure" an object has—regularity, patterns, rules, etc.—the more easily it can be described by a short program and the lower its Kolmogorov complexity. Kolmogorov complexity therefore is deeply rooted in the idea of compression. For instance, a sequence with repeating patterns or a dataset that spans a low-dimensional subspace can be significantly compressed relative to its original size, and this results in low Kolmogorov complexity. In contrast, a random string devoid of any structure cannot be compressed at all and must in effect be "hard-coded", making its Kolmogorov complexity equal to its original size in bits.

While powerful, Kolmogorov complexity has certain limitations. First and foremost, Kolmogorov is intractable to compute exactly because it requires a brute force search over an exponentially large space of possible programs. It is therefore often of conceptual rather than practical value, although it can nevertheless be upper-bounded using more efficient compression strategies. Second, Kolmogorov complexity depends on the programming language of choice. For instance, if a programming language has a built-in primitive for the object being encoded, Kolmogorov complexity is trivially small. This concern, however, is often overblown: given any two Turing-complete programming languages, the difference in Kolmogorov complexity that they assign to an object is upper-bounded by a constant that is independent of the object itself, because any Turing-complete programming language can simulate another (Grünwald & Vitányi, 2003; Fortnow, 2000). In practice, we can simply consider "reasonable" Turing-complete programming languages that don't contain arbitrary object-specific primitives, in which case this simulation constant will be relatively small and the particular programming language of choice will have little effect. Finally, Kolmogorov complexity is only defined for discrete objects because no terminating program can output a continuous number with infinite precision. This concern is also less consequential in practice, because we can always represent continuous objects using finite (e.g., floating-point) precision.

**Important properties for machine learning.** In ML, we are often concerned with datasets and probabilistic models. Kolmogorov complexity relates to these two concepts in several interesting ways. First, we can ask about the Kolmogorov complexity of a finite dataset $X = (x_1, ..., x_n)$ where each sample is drawn *iid* from a distribution $p(x)$. It turns out that if

we have access to the true distribution $p(x)$, optimal algorithms such as arithmetic coding (Witten et al., 1987) can encode each sample using only $\log_2 p(x_i)$ bits. Intuitively, this is because samples that occur more frequently can be encoded using shorter codes in order to achieve an overall better compression. We thus have that:

$$K(X|p) = -\sum_{i=1}^{n} \log_2 p(x_i). \tag{9}$$

If instead of access to the true distribution $p(x)$ we only have a probabilistic model of the data $p_\theta(x)$, we have that:

$$K(X|p_\theta) \leq -\sum_{i=1}^{n} \log_2 p_\theta(x_i), \tag{10}$$

where we have equality when $p_\theta = p$. This insight is significant. Notice that $-\sum_{i=1}^{n} \log_2 p_\theta(x_i)$ is the negative log-likelihood of the data under the model, which is a common loss function used in ML. This tells us that models with lower error better compress their data, and directly relates Kolmogorov complexity to optimization in ML. However, what if we do not have a model? What is the Kolmogorov complexity of the data itself? Intuitively, if the dataset is sufficiently large, the optimal method for encoding it should be to first specify a model and then encode the data using that model as in Equation (10). Specifically, using identities in Fortnow (2000), we have:

$$K(X) \leq K(X|p_\theta) + K(p_\theta). \tag{11}$$

This encoding scheme on the RHS is referred to as a 2-part code (Grünwald, 2007). We have equality when the model's description length and error are jointly minimized, which occurs when the model $p_\theta(x)$ is equivalent to the true distribution $p(x)$:

$$K(X) = \arg\min_{p_\theta} K(X|p_\theta) + K(p_\theta) = \arg\min_{p_\theta} -\sum_{i=1}^{n} \log_2 p_\theta(x_i) + K(p_\theta) \tag{12}$$

$$= K(X|p) + K(p) = -\sum_{i=1}^{n} \log_2 p(x_i) + K(p). \tag{13}$$

Again, we can draw important connections to ML. Equation (11) says that the Kolmogorov complexity of a dataset is upper-bounded by the a model's error and complexity. In addition, Equations (12) and (13) tell us that the simplest model that explains the data is most likely to be the true one, which draws a theoretical link between compression, maximum likelihood training, model complexity, and generalization (Goldblum et al., 2024).

**Relation to Shannon information.** In Shannon information theory (Shannon, 2001), the notion of information quantity is entropy. Given a random variable $X \sim p(x)$, entropy is defined as: $H(X) = \mathbb{E}_{x \sim p(x)} - \log_2(p(x))$. Notice that the $-\log_2(p(x))$ inside the expectation is equal the quantity inside the sum of Equation (9), which specified the minimum number of bits needed to encode a sample from a dataset given the distribution that sample was drawn from. This is no accident: entropy can be seen as the average number of bits needed to compress events from a distribution using an optimal encoding scheme when the distribution $p(x)$ is known. If we simply sum these bits for a finite number of samples instead of taking an expectation, we get exactly $K(X|p)$ as defined in Equation (9).

As we have seen, though, the assumption about a known distribution $p(x)$, need not be made in the Kolmogorov complexity framework. In this sense, Kolmogorov complexity is a strict generalization of Shannon information theory: $K(X)$ as defined in Equation (13) is equivalent to summed entropy plus the complexity of the distribution $p(x)$, which is unknown and needs to be encoded. In the Shannon framework, it is difficult to derive a meaningful notion for the information quantity in the distribution $p(x)$ because it is an individual object—a function, in particular—and Shannon information is only defined for random variables (Grünwald & Vitányi, 2003). A second drawback of Shannon information is that entropy is a measure of statistical determinability of states; information is fully determined by the probability distribution on states and unrelated to the representation, structure, or content of the individual states themselves (Grünwald & Vitányi, 2003). For this current work, we require a notion of complexity that can account for representations and functions, making Kolmogorov complexity better suited to the task.

# B. Prequential coding and compression without a known learning algorithm

When introducing the relationship between prequential coding and optimal compression in Equation (4), we mentioned that a key assumption is that the learning algorithm $T$ is known. In reality, then, we have that:

$$K(D) \leq K(D, p_\theta) \tag{14}$$
$$= K(D|T) + K(T) \tag{15}$$
$$\leq L_{preq}(D; T) + K(T) \tag{16}$$
$$\implies L_{preq}(D; T) + K(T) \geq K(D), \tag{17}$$

where inequality Equation (14) appears because compressing additional objects can only take more bits, and inequality Equation (16) comes from the fact that prequential coding is not necessarily the optimal way to compress a dataset given a learning algorithm. If the learning algorithm is a short program like SGD, however, then $K(T) \approx 0$ and $L_{preq}(D; T)$ is an upper-bound on $K(D)$. For simple learning algorithms, then, Equation (4) holds.

# C. Experiment and task details

In this section, we provide additional experimental details, including a comprehensive overview of the model architectures and hyperparameters used during training. All experiments were run on GPUs with at least 32 GB of RAM, and each took less than 1 day to run on a single NVIDIA V100 with all seeds stated in figure captions.

## C.1. Meta-learner architectures

We considered different architectures which exhibit ICL to study and compare their ability to minimize prequential code length (Section 3.3). Each architecture described here parameterizes the meta-learner $T_\phi$.

**Transformer with bottleneck.** We use a standard causal decoder-only Transformer with 4 layers, 4 attention heads, 256 latent dimensions and a feed-forward network with 512 dimensions. Additionally, it has linear projection that bottlenecks the Transformer to 128 dimension. A 5-layer MLP with RELU activations and 256 latent dimensions is used as a separate prediction head.

The Transformer takes a dataset $D$ as input in the format $[x_1, y_1], [x_2, y_2], \dots, [x_n, y_n]$ (where $x_i$ and $y_i$ are concatenated and each $[\cdot]$ is a token) and computes $T_\phi(D_{1:t-1})$ for each context size starting from 1 to $n-1$. The computation of $T_\phi(D_{1:t-1})$ is based on the encoding of the $t$-th token, which attends only to tokens that appear to the left of $[x_t, y_t]$ and itself. Information leakage from future tokens is prevented using a causal mask. After computing $T_\phi(D_{1:t-1})$, we concatenate it with $x_t$ (i.e., $[T_\phi(D_{1:t-1}), x_t]$) and pass this combined input to an MLP prediction head to predict the next $y$-token.

**Transformer without bottleneck.** We use a custom encoder-decoder Transformer with 4 layers, 4 attention heads, 256 latent dimensions and a feed-forward network with 512 dimensions. Also, in contrast to the previous architecture we don't use a separate prediction head.

To allow for parallel processing at each position $x$ without leaking information about the corresponding $y$ in a model without bottleneck, we augment a standard Transformer architecture in the following manner. It considers two sets of tokens, namely (a) $D$ in the format $[0, 0], [x_1, y_1], [x_2, y_2], \dots, [x_n, y_n]$ (where $x_i$ and $y_i$ are concatenated for each token), and (b) $X$ in the format $[x_1], [x_2], \dots, [x_n]$ (where each token only has $x$ information). Note that $[\cdot]$ describes a token, and the first token in $D$ represents an empty context.

Each layer of this Transformer performs the following attention procedures:

$$X^{(l)} = \text{Attention}\left(\text{Query} = X^{(l-1)}, \text{Key} = D^{(l-1)}, \text{Value} = D^{(l-1)}, \text{Mask} = \mathcal{M}^X\right) \tag{18}$$

$$D^{(l)} = \text{Attention}\left(\text{Query} = D^{(l-1)}, \text{Key} = D^{(l-1)}, \text{Value} = D^{(l-1)}, \text{Mask} = \mathcal{M}^D\right) \tag{19}$$

where $\mathcal{M}^X$ ensures that $X_t^{(l-1)}$ can only attend to $D_{1:t-1}^{(l-1)}$ and $\mathcal{M}^D$ ensures that $D_t^{(l-1)}$ can only attend to $D_{1:t}^{(l-1)}$. Both $X^{(l)}$ and $D^{(l)}$ go through a residual feed-forward network after the attention operations.

Note that the above operation achieves two distinct properties: (a) it prevents the token $[x_t]$ from accessing information about $y_t$ while allowing access to all $x_{1:t-1}$ and $y_{1:t-1}$ in making the corresponding prediction, and (b) akin to standard Transformers the $[x_t, y_t]$ token can attend to $x_{1:t}$ and $y_{1:t}$.

**Mamba.**  We experiment with two state-space model (SSM) architectures, Mamba 1 and Mamba 2, both composed of 4 layers, 256 latent dimensions, state dimensions 8, and local convolution dimension of 4. Additionally, each layer includes a gated MLP with 256 latent dimensions. Similar, to the Transformer with bottleneck, the prediction model is a 5-layer MLP with RELU activations and 256 latent dimensions is used as a separate prediction head.

The SSM takes a dataset $D$ as input in the format $[x_1, y_1], [x_2, y_2], \ldots, [x_n, y_n]$ (where $x_i$ and $y_i$ are concatenated and each $[\cdot]$ is a token). For each context of size $t-1$, we compute the $T_\phi(D_{1:t-1})$ which is a vector that represents the parameters of the output model obtained after processing the first $t-1$ data points. After computing $T_\phi(D_{1:t-1})$, we concatenate it with $x_t$ (i.e., $[T_\phi(D_{1:t-1}), x_t]$) and pass this combined input to an MLP prediction head to predict the next $y$-token.

## C.2. Meta-training and evaluation setup

In this section, we outline the complete set of hyperparameters and configurations used across different training objectives and model architectures in our experiments.

**In-context learner (prequential and train-risk).**  We trained both the Transformer-based meta-learners (with and without bottleneck) for 50 epochs and the Mamba-based meta-learners for 120 epochs. All results were averaged across 5 different random seeds to mitigate the effect of randomness in the pipeline. The training was conducted on a meta-dataset consisting of 10,000 tasks, each with 1,000 data points that serve as context. We used the Adam optimizer (Kingma & Ba, 2015) with a learning rate of $\eta = 0.0001$ and a batch size of 256, without any early stopping. After meta-training, we evaluated the learners on a distinct meta-dataset of 100 tasks, each with 1,000 data points.

**Gradient based learner.**  Since gradient-based learner are off-the-shelf learning algorithms which don't require meta-training. The prediction model used is a 5-layers MLP with RELU activations and latent dimensions of 64 or 256 depending on the complexity of the task. We used a meta-dataset of 10000 tasks (with 2000 data points each) split into training (80%) and validation (20%). At each step of prequential coding, we train and evaluate a model by randomly sampling a dataset of fixed size across each of the tasks, starting from 20 to 2000 datapoints. We used an early stopping criteria with minimum loss delta of 0.001 and patience of 10 epochs to avoid overfitting. On each of them, the prediction model was fit using the Adam optimizer (Kingma & Ba, 2015) with a learning rate of $\eta = 0.0001$ and a batch size of 64. All results were averaged across 15 different random seeds.

## C.3. Visualization of the model inferred in-context

For a given context length, next-token loss serves as a quantitative proxy for the overfitting behavior of a learned prediction function. Beyond that metric, in experiments generating Figure 3, we visualized the kinds of models inferred by ICL on simple regression problems in a way that lets us clearly inspect their complexities. The tasks consist of noisy polynomial regression problems of the form:

$$f(x) = \sum_{i=1}^{N} \alpha_i C_i(x) + \epsilon,$$

where $C_i$ is the $i$-th Chebyshev polynomial of the first kind and $\epsilon \sim \mathcal{N}(0, \sigma^2)$. We use this basis of polynomials instead of the usual canonical basis for both data generation and the hypothesis class of our learned predictor, due to the favorable numerical properties (e.g., orthogonality) of the Chebyshev basis. Specifically, when used to generate data, Chebyshev polynomials offer better coverage of different functional behaviors; and when used for function approximation, they lead to smaller approximation error and faster convergence.

We train prequential ICL and train-risk ICL for 400 epochs on a meta-distribution of 20000 functions of degree up to $k$. We experiment with both $k = 1$ (linear functions) and $k = 3$ (cubic functions). We use a bottlenecked model (Section C.1)

to infer the Chebyshev polynomial coefficients $\alpha \in \mathbb{R}^N$ of the unknown function $f$, where we make $N \gg k$ so that the in-context models have the capacity necessary to overfit the training data. These inferred coefficients are then passed to a parameter-less prediction head that, given a query input $x_q$, computes:

$$f(x_q, \alpha) = \sum_{i=1}^{N} \alpha_i C_i(x).$$

Restricting the regression tasks to functions of degree $k \ll N$ allows us to compare learners in the over-parameterized regime, which is the most common settings in modern machine learning. Our results, shown in Figure 3, align with the trends observed in Figure 2, confirming that train-risk ICL more frequently exploits high-degree components (i.e., degrees strictly greater than $k$) to overfit in-context training data, compared to prequential ICL which fits simpler models closer to the ground-truth data-generating processes.

### C.4. Pretrained LLM on Mastermind

As described in Section 3.4, we evaluate the performance of a pretrained LLM on the Mastermind task using one of the latest OpenAI models GPT-4 (i.e., gpt-4o). To query the model, we used the OpenAI API with a `temperature` of 0, ensuring that the outputs are deterministic. Along with the responses, we also obtained the log probabilities using the API for calculating the prediction error with respect to each query. This was possible using `logprobs` (boolean) and `top_k_logprobs` (integer) attributes in the API that returns log probabilities for each token in the response and the $k$ tokens with the top log probabilities corresponding to each token in response. By using a structured prompting technique and a retry mechanism (up to 10 retries in case of failure to adhere to the required output format), we were able to consistently obtain appropriate responses to our queries. An example prompt, which includes the task description, context, and the query, is provided below. To calculate the prequential code length, we iteratively query novel examples with an increasing number of in-context examples and obtain the prediction errors. This process emulates the prequential ICL objective.

---

**Example Prompt**

```
I have a secret code in mind.  It's a 8-digit code with each digit ranging
between 0 and 5.  I'll give you a couple example guesses, and for each guess
I'll tell you two numbers:

- First number:  the number of correct correct digits at their correct
position.  - Second number:  the number of correct digits, which aren't
necessarily in the correct position.

Here's a demo to show you what a guess and response would look like.
Imagine my secret code was:
0 5 2 1 3 4 2 4
And imagine the guess I presented you was:
0 2 1 1 0 2 0 4
Then, the response would be:
3 5

The response is the way it is because the first, forth and last digit were
in the correct place (first response number is therefore 3) and additionally
the second and sixth digit were in the guess but at the wrong position
(second response number is therefore 5).

The game is about to start.  I'll present you with a series of guesses and
their responses.  Finally, I will present you with a new guess, and you'll
have to predict the correct response.  Make sure your response is formatted
the same way as in the examples (i.e., with 2 digits between 0-8, separated
by a space).  Let's begin.
```

```
----------------------
Guess:  4 2 1 3 4 0 0 5
Response:  3 7

Guess:  1 1 4 3 5 5 0 1
Response:  2 5

Guess:  3 0 2 2 0 5 3 4
Response:  2 6

Guess:  0 2 5 0 4 2 0 1
Response:  1 5

Guess:  4 1 3 2 5 4 2 3
Response:  ?  ?
----------

What do you think the response is for this final guess?  Make sure to reply
with just 2 digits between 0-8, separated by a single space character.
```

### C.5. Hidden Markov Model task

A prominent theory for why ICL emerges from the next-token prediction objective of LLMs is that sequences $x_{1:n}$ in the pretraining dataset (e.g. large corpuses of text) can be interpreted as implicitly being sampled from a latent variable generative model $Q(x_{1:n} \mid \tau)$ where $\tau$ are some abstract *concepts* underlying samples (Chan et al., 2022; Xie et al., 2022). $\tau$ can range from abstract *style* attributes in natural language (Xie et al., 2022) to *task parameters* such as the teacher weight matrix in linear regression ICL task (Von Oswald et al., 2023); the important part is that some latent variables can be inferred from the context and subsequently aid prediction. ICL would then emerge as the ability of performing implicit Bayesian inference (i.e. learn from the context) in order to predict $x_t$ :

$$Q(x_t \mid x_{<t}) = \sum_{\tau} \underbrace{Q(x_t \mid x_{<t}, \tau)}_{\text{Condition on the latent}} \underbrace{Q(\tau \mid x_{<t})}_{\text{Infer latent}} \tag{20}$$

We propose to leverage this conceptual framework to devise a novel generation procedure for synthetic LLM pretraining datasets. The general idea is to design a family of sequence models $Q_{\tau}(x_{1:n})$ parameterized by task latents $\tau$, leading to the latent variable generative distribution

$$Q(x_{1:n} \mid \tau) = Q_{\tau}(x_{1:n}).$$

Specifically, we use Hidden Markov Models (HMMs) as the sequences models, and we parameterize the HMMs $Q_{\tau}(x_{1:n})$ with parameters $f_{\xi}(\tau) = \psi_{\tau}$. We use this function $f$ to introduce hyper-parameters $\xi$ which define the whole family of sequence models; i.e. the dataset. Below, we define in details a specific *ad-hoc* function $f_{\xi}(\tau)$ which generates a family of HMM where each member share non-trivial structure.

#### C.5.1. DETAILED DESCRIPTION OF THE GENERATIVE PROCESS

A HMM defines a probability distribution over sequences of *observations* $x_i \in \mathcal{X}$ with a discrete-time probabilistic process over *hidden states* $z_i \in \mathcal{Z}$ paired with a mapping $\mathcal{Z} \to \mathcal{X}$. Both $\mathcal{X}$ and $\mathcal{Z}$ are discrete sets. The hidden process is defined by an initial state distribution $\pi(z)$ and a transition matrix $A \in \mathbb{R}^{|\mathcal{Z}| \times |\mathcal{Z}|}$ such that

$$Q(z_i | z_j) = A_{ji}$$

Lastly, the mapping between states and observations is governed by the emission matrix $B \in \mathbb{R}^{|\mathcal{Z}| \times |\mathcal{X}|}$ such that

$$Q(x_j | z_i) = B_{ji}$$

In the rest of the section, we will explicitly define how $f_{\boldsymbol{\xi}}(\boldsymbol{\tau})$ generates $\psi_{\boldsymbol{\tau}} = (\pi^\tau, A^\tau, b^\tau)$. We first give a high level description.

The *hyper-parameters* $\boldsymbol{\xi}$ will define a number of building blocks which will be used to create the transition and emission matrix of all HMMs. Then $\boldsymbol{\tau}$ will specify a specific way to combine and manipulate these building blocks to instantiate a specific HMM $Q_{\boldsymbol{\tau}}$. For the transition matrix $A^\tau$, the building blocks are pre-defined cycles; which are combined, flipped and accelerated based on $\boldsymbol{\tau}$. For the emission matrix $B^\tau$, the building blocks are groups of sub-emission matrices which each only affect a subset of $|\mathcal{X}|$; which are combined and possibly internal shifted based on $\boldsymbol{\tau}$. Overall, we will have

$$\begin{aligned}
\boldsymbol{\xi} = (&\text{N\_BASE\_CYCLES}, \text{N\_BASE\_SPEEDS}, \text{N\_CYCLE\_FAMILIES}, \\
&\text{N\_GROUP\_PER\_FAMILY}, \text{N\_FAMILY\_SPEEDS}, \text{N\_EMISSION\_GROUPS}, \\
&\text{N\_EMISSION\_PER\_GROUP}, \text{N\_EMISSION\_SHIFT})
\end{aligned}$$

and

$$\begin{aligned}
\boldsymbol{\tau} = (&\text{BASE\_ID}, \text{BASE\_SPEED}, \text{FAMILIES\_IDs}, \\
&\text{FAMILIES\_SPEED}, \text{EMISSION\_IDs}, \text{EMISSION\_SHIFT})
\end{aligned}$$

We will refer to the dimensions of $\boldsymbol{\xi}$, $\boldsymbol{\tau}$ as $\xi_i$, $\tau_i$ to avoid clutter and discuss further details below.

**Transition matrix $A^\tau$.** We define a cycle as sequence of hidden states $\boldsymbol{c} = (c_0, \ldots, c_{|c|-1})$, $c_i \in \mathcal{Z}$, and the following manipulation functions

$$\text{DIR}(\boldsymbol{c}, k) = \begin{cases} (c_0, c_{|c|-1}, \ldots, c_1) & \text{if } k = 1 \\ \boldsymbol{c} & \text{otherwise.} \end{cases}$$

$$\text{SPEED}(\boldsymbol{c}, k) = (c_0, c_{k(\text{mod } |c|)}, c_{2k(\text{mod } |c|)}, \ldots)$$

In words, $\text{SPEED}(\boldsymbol{c}, k)$ changes the speed at which the cycle is traversed and $\text{DIR}(\boldsymbol{c}, k)$ change its direction. We finally define the transition matrix $\mathcal{T}(\boldsymbol{c})$ associated with cycle $\boldsymbol{c}$ such that

$$\mathcal{T}(\boldsymbol{c})_{ij} = \begin{cases} 1 & \text{if } \exists k < n \text{ s.t } (i, j) = (c_k, c_{k+1(\text{mod } n)}) \\ 0 & \text{otherwise.} \end{cases}$$

Initially, we randomly generate $\xi_0$ *base cycles* $\boldsymbol{b}_i$ which go through all states $z_i$. Further, we initialize $\xi_2$ families of $\xi_3$ groups of cycles $\boldsymbol{g}_j^i$, $i \in [\xi_1]$, $j \in [\xi_2]$. Each HMM's transition matrix is then built from these "building blocks" cycles. Specifically,

$$A^\tau = \mathcal{T}(\text{SPEED}(\text{DIR}(\boldsymbol{b}_{\tau_0}, \tau_1), \tau_2)) + \sum_{i=1}^{\xi_2} \tau_{4,i} \sum_{j=1}^{\xi_3} \cdot \mathcal{T}(\text{SPEED}(\text{DIR}(\boldsymbol{g}_j^i, \tau_5), \tau_6))$$

In words, each transition matrix is made of a) one of $\xi_0$ base cycle, possibly sped up and flipped and b) $\xi_2$ groups of smaller cycles (each from a pool of $\xi_3$ groups), possibly sped up and flipped. The number of possible speeds for the base cycle is defined by $\xi_1$. For the cycle families, it is defined by $\xi_4$

**Emission matrix $B^\tau$.** We separate the states $z \in \mathcal{Z}$ in $\xi_5$ groups $\boldsymbol{h}_i \subset \mathcal{Z}$ and for each group we initialize $\xi_6$ sub-emission matrices $H_j^i \in \mathbb{R}^{|\boldsymbol{h}_i| \times |\mathcal{Z}|}$. Then, we define the manipulation function $\text{SHIFT}(H, k)$ which applies a circular shift of $k$ to the indices of the matrix. Finally, we have

$$B^\tau = \sum_{i=1}^{\xi_5} \text{SHIFT}(B_{\tau_7, i}^i, \tau_8)$$

In words, each emission matrix is made of $\xi_5$ possibly overlapping sub-emission matrix, each picked from a pool of $\xi_6$ unique ones. The number of possible shifts is $\xi_7$.

**Initial distribution.** We always use the uniform distribution.

C.5.2. HMM HYPER-PARAMETERS

For experiments in this paper, we use $|\mathcal{X}| = 50$ and $|\mathcal{Z}| = 20$. The hyper-parameters of $f$, $\boldsymbol{\xi}$, are given in Table C.1. This results in a total of 512 different transition matrices and 24 different emission matrices, for a total of 12,228 different HMMs. We show results averaged from 5 different seed.

| | |
|---|---|
| N_BASE_CYCLES ($\xi_0$) | 4 |
| N_BASE_SPEEDS ($\xi_1$) | 2 |
| N_CYCLE_FAMILIES ($\xi_2$) | 3 |
| N_GROUP_PER_FAMILY ($\xi_3$) | 2 |
| N_FAMILY_SPEEDS ($\xi_4$) | 2 |
| N_EMISSION_GROUPS ($\xi_5$) | 3 |
| N_EMISSION_PER_GROUP ($\xi_6$) | 2 |
| N_EMISSION_SHIFT ($\xi_7$) | 3 |

*Table C.1.* **HMM dataset hyper-parameters**

C.5.3. TRAINING

We train on 200,000 trajectories from 60,000 unique compositions of latent variables, and evaluate on 50,000 trajectories from 13,728 held out compositions of latent variables. Training consists on next-token prediction with a cross-entropy loss. We use the same model and training hyperparameters as in Section C.1 and Section C.2.

## D. Effect of task difficulty on prequential code length

In Section 3.1 Figure 2, we found that a meta-learned in-context learner trained to minimize prequential code length (prequential ICL) was better able to generalize than one that only minimized training error (train-risk ICL). We further noted that the gap in generalization error between these two learners was greater in low-data regimes, and that the gap extended further as a function of task difficulty (i.e., more in-context data was required to close the gap going from linear regression, to sinusoid regression, to Mastermind). This result is predicted by our theory relating ICL to Occam's razor. A complex task requires the algorithm to learn more complex functions to successfully minimize train risk. However, learning more complex functions with very limited data leads to overfitting, which is the basis for our hypothesis that as task complexity increases, simple predictors learned by minimizing prequential code length enjoy a bigger advantage over predictors learned by minimizing train risk.

To investigate the effect of task difficulty more systematically in this section, we fix the underlying meta-dataset (sinusoid regression tasks) and vary the dimensionality of the input data $dim(x)$. We plot our results in Figure D.1, showing the difference in generalization error between train-risk ICL learners and prequential ICL learners. As expected, ask task difficulty increases, this generalization gap extends further, and the train-risk learners must observe more data in-context in order to close it.

## E. Effect of SGD regularization on prequential code length

Regularization techniques are widely used for gradient-based learners to prevent over-fitted solutions. In this experiment we fit prediction models considering different regularization techniques, namely early-stopping combined with validation data, and weight-decay (L2 regularization). The results are presented in Figure E.1.Experiments with early-stopping halt training when the validation loss does not decrease by more than $1e - 4$ over 10 consecutive steps. Experiments with weight-decay consider a regularization parameter $\lambda \in \{0.05, 0.005\}$ and were trained for 1000 epochs. The prediction models used are 5-layers MLPs with RELU activations and latent dimensions of 64. The different prediction models were fit using an Adam optimizer (Kingma & Ba, 2015) with a learning rate of $\eta = 0.0001$ and a batch size of 64. All results were averaged across 15 different random seeds.

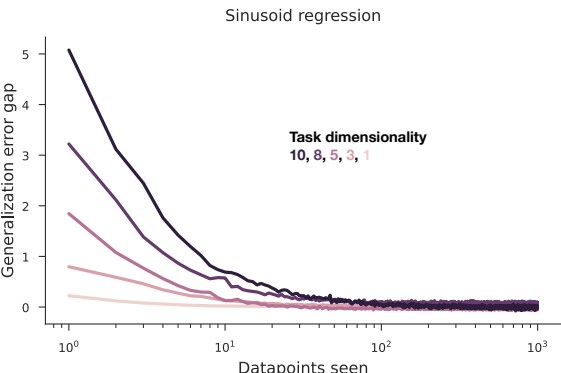

*Figure D.1.* **Comparison of gap between prequential ICL and train-risk ICL as a function of task difficulty.** Figure shows the difference in average prequential coding curves (i.e., generalization error for train-risk ICL − generalization error for prequential ICL) for sinusoid regression tasks of increasing input dimensionality. Error is measured using MSE. Error bars show standard error across 5 seeds. For all task dimensionalities, the performance gap is positive: ICL from next-token prediction objectives (prequential ICL) yields lower prequential code lengths than ICL from past-token prediction objectives (train-risk ICL), with greater effects in low-data regimes. This gap in generalization error increases with task dimensionality, demonstrating that learners which minimize prequential code length generalize better in virtue of fitting simpler models, and that these simpler models are most important when generalization is difficult (i.e., when the task dimensionality is large relative to the amount of training data observed).

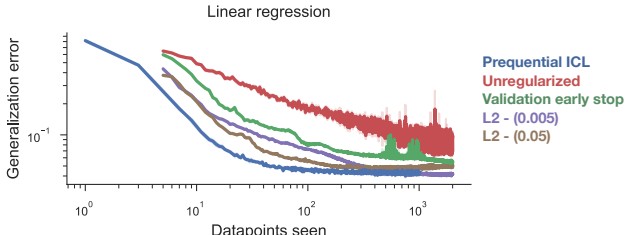

*Figure E.1.* **Experimental results comparing different regularization techniques.** Figure show average prequential coding curves obtained using both unregularized and regularized Adam optimizers on a linear regression task. Regularized learners exhibit better compression rate (i.e. lower PCL), which implies a stronger incentive toward simple models according to our theory. This experiment confirms the claim that regularization techniques serve as indirect Occam's aligned methods to learn simple models. Analogous to the meta-learning setting, PCL could be minimized with respect to the hyperparameters of the regularization technique.

## F. Hidden-Markov-Model task comparisons to ICL with a train-risk objective and SGD

In Section 3.1, we only considered *iid* tasks so that we could make fair comparisons to learners that minimize training error under *iid* assumptions (i.e., train-risk ICL and SGD). In this section, we nevertheless attempt to make these comparisons on our synthetically-generated nonstationary data from our Hidden Markov Models (HMMs). See Appendix C for further details on this data.

By default, our HMM tasks are not amenable to learners that minimize train-risk. This is because in our theory, we see individual context tokens as "datapoints" that are processed separately by a train-risk model to minimize prediction error. However, on a nonstationary sequence dataset, the learnable structure is about the relationship *between* datapoints. While this sort of structure is ordinarily learned by minimizing next-token prediction error, this amounts to prequential ICL. A train-risk baseline would somehow need to be trained to minimize error on previously observed tokens in the context, and there is no way to query the model on such tokens at training time while evaluating its predictions on the next token at inference time.

To make our HMM tasks amenable to learners that minimize train-risk, we applied a simple transformation that turned the sequence datasets into supervised learning problems. Call a sequence of HMM observations $D = (y_1, ..., y_k)$, where the subscript denotes the time index. We transform $D$ into a supervised learning dataset $D' = \{(x_1 = 1, y_1), ..., (x_k = k, y_k)\}$, where $y_i$ is a model observation and $x_i$ is the integer time index of that observation. In this way, we can evaluate train-risk

ICL and SGD in the same way as for our *iid* supervised datasets, and measure prequential code length by querying the resulting models on the next timepoint $x_{k+1}$. For fair comparison, prequential ICL was also trained and evaluated used the same supervised version of the HMM task. As a technical note, we embed each $x_i$ using relative positional encodings (Shaw et al., 2018).

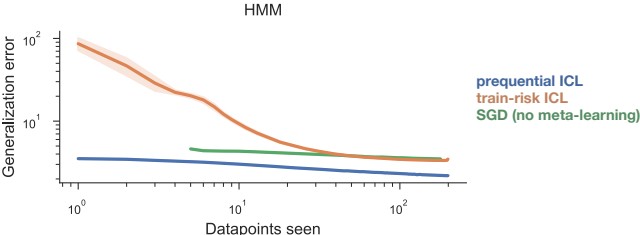

*Figure F.1.* **Comparisons between prequential ICL, train-risk ICL, and SGD on nonstationary HMM data.** Average prequential coding curve for a meta-dataset sampled from HMMs, which is the mean prediction error on the next unobserved token (generalization error, y-axis) given observed contexts of increasing length (datapoints seen, x-axis). The area underneath these curves corresponds to prequential code length. Error is measured using cross-entropy. Error bars show standard error across seeds (5 for ICL, 15 for SGD).

Our results are shown in Figure F.1, where we use the same models and training setup as in Section 3.1. We find that prequential ICL, as expected, minimizes prequential code length well and generalizes at all context lengths. This shows how prequential coding through ICL can fit simple models that compress not only *iid*, but also nonstationary data due to the quickly-adapting online nature of the learning algorithm. In contrast, both train-risk ICL and SGD fail to generalize at every context length, and are unable to predict the next token in a sequence at inference time after having only been trained to fit the earlier part of the sequence.

## G. Effect of sequence length used at pretraining for HMM task: Full vs. Partial

As argued in Section 2.4, an in-context learner behaves as an effective online compression algorithm known as prequential coding. The fact that it implements such procedure lies in the fact that it is tasked to iteratively make predictions on novel data given increasing numbers of datapoints in-context, starting from an empty context.

Exactly as we did with train-risk ICL, we could have come up with a different training objective, such that the in-context learner trained on it no longer implements prequential coding. One possibility is to train an in-context learner to predict only on the last half of the context instead of the full context. Note that in both cases, the learner still observes the full context, but gradients are only computed and backpropagated for predictions made on the second half. This adjustment is made to mimic the standard way of training of LLMs, where predictions are primarily made under large-context conditions; even though early tokens in a sequence initially have shorter contexts, the majority of training steps occur once the context window is sufficiently filled. While this training scheme does not correspond to any known coding algorithm for data compression, we conjecture that such learner has a weaker incentive toward compression and should therefore yield worse generalization performance than its prequential ICL counterpart.

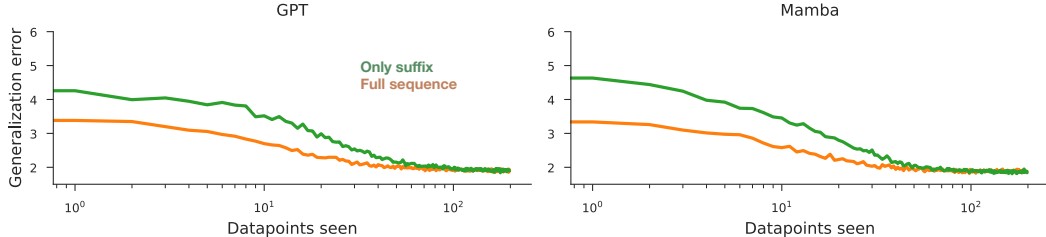

*Figure G.1.* **Impact of partial-context training on next-token prediction in HMM data.** Mean next-token prediction error (generalization error, y-axis) as a function of observed context length (x-axis), comparing models meta-trained trained to predict full sequences versus only the latter half of sequences. Results are shown for two architectures: a GPT-2-style transformer and Mamba. For contexts longer than 100 tokens (half the maximum sequence length), there is no statistically significant difference in generalization error between the two training regimes, suggesting that the In-Context learner trained the prequential way does not learn simpler model in this setting.

As Figure G.1 shows, the model pre-trained on sequences for which no auto-regressive gradient was used for the first half of the sequence (shown as "only suffix") shows worse generalization than the model training to minimize the full prequential coding curve. As expected, this discrepancy is stronger in low data regimes and the gap between the two models shrinks as testing sequence length grows. Future experiments could further explore generalization properties of pre-trained learners based on accessible sequence length and on task *iid* requirements.

## H. Advantages over the Bayesian perspective

The Bayes-optimal prediction perspective of ICL and meta-learning says that by meta-training on some set of tasks $\mathscr{D}$, the learner infers some prior over latent task variables—or, equivalently, a prior over models—$p(p_\theta|\mathscr{D})$. On some novel task $D$, the learner then infers a posterior over models that both explain the training data (i.e., assign it a high likelihood) and are consistent with the prior: $p_\mathscr{D}(p_\theta|D) = p(D|p_\theta)p(p_\theta|\mathscr{D})/Z$, where $Z$ is a normalizing constant. According to the theory, subsequent predictions are then done through implicit Bayesian averaging under this posterior model distribution.

Crucial differences in our theory are that $\mathscr{D}$ does not need to be drawn from a well-defined distribution over tasks for us to reason about the meta-learning problem (the Kolmogorov framework does not require this) and minimizing $\sum_{i=1}^{M} L_{preq}(D^i; T_\phi) \geq \sum_{i=1}^{M} K(D^i|T_\phi)$ need not induce a prior probability distribution over models given $\mathscr{D}$ (although it can if this is the best way to compress the meta-dataset using the meta-learner $T_\phi$). As a result, our theory generalizes the Bayesian perspective.

To see why these generalizations provide value, consider where the prior in the Bayesian framework $p(p_\theta|\mathscr{D})$ comes from. This prior is not defined explicitly in the ICL framework; instead, it is implicitly defined based on $\mathscr{D}$, the implicit *initial* prior $p(p_\theta)$, and the implicit inference machinery that approximates $p(p_\theta|\mathscr{D}) = p(\mathscr{D}|p_\theta)p(\theta)/Z$. All of these implicit components make any meaningful analysis difficult, since it is difficult to characterize them. However, these implicit components are all intrinsic properties of the meta-learning algorithm (the meta-learner's architecture, the meta-objective, etc.), which we *do* have *explicit* control over. Our theory only makes reference to this meta-learner $T_\phi$ and the description length of data under it $L_{preq}(D^i; T_\phi) \geq K(D)$, rather than to objects that are only implicitly defined (and never known). As such, we argue that our theory is more amenable to analysis and provides more explanatory value.

For example, in the Kolmogorov framework that we have proposed, it is easy to see how ICL might in some cases generalize to a novel dataset $D$ that is entirely out-of-domain with respect to $\mathscr{D}$. Perhaps, for instance, the tasks have compositional structure and $T_\phi$ has some inductive biases for compositional generalization. In contrast, it is far more difficult to find a good explanation for such a phenomenon in the Bayesian framework. The explanation would have to be in terms of some implicit initial prior $p(p_\theta)$ (which we never defined) and the subsequent prior $p(p_\theta|\mathscr{D})$ that it induced, which can easily lead to just-so stories of the form "generalization here must have been possible because $p(p_\theta)$ had the right kind of structure". However, this sort of post-hoc rationale could be used to explain *any* outcome (positive or negative), and is therefore problematic as a scientific explanation (Deutsch, 2012).

Another problem with the Bayesian perspective is that its predictions do not always hold in practice. Notably, Raventós et al. (2024) found that when the diversity in pretraining tasks is sufficiently large, solutions emerge that are *not* consistent with a Bayes-optimal predictor that uses the pretraining task distribution as its prior. Instead, the solution is consistent with a much broader prior, which allows the learner to adapt to novel tasks that are outside of the pretraining task distribution. Our theory, in contrast, permits explanations for this phenomenon. For instance, perhaps that model used to parameterize $T_\phi$ had insufficient capacity to encode a diverse (and potentially complex) prior over tasks, and instead learned a simpler approximation with more broad coverage over a larger space of tasks.

