# OpenReview forum: "In-Context Learning and Occam's Razor"
_ICML.cc/2025/Conference — ICML 2025 poster_

### Official Review · Reviewer_tMvj · 2025-03-14

**Overall Recommendation:** 1

**Summary:**

The paper studies ICL and tries to relate ICL to prequential coding length through the scope of Kolmogorov complexity.

**Claims And Evidence:**

There're some vague claims stated by the authors, see questions part.

**Essential References Not Discussed:**

Not any that I know of.

**Experimental Designs Or Analyses:**

The experimental designs are confusing, see questions part.

**Methods And Evaluation Criteria:**

Proposed methods and evaluation criteria is confusing.

**Other Comments Or Suggestions:**

1. line 153 "Figure 1b" should be "Figure 1".

**Other Strengths And Weaknesses:**

### Strengths
1. The paper studies ICL, which is an important ML topic.

### Weaknesses
1. The paper claims to provide a theory on ICL, but no formal theorems or statements can be found in the paper.
2. The relationship with ICL is vague, the next-token prediction is not the essence of ICL, it is the training paradigm of all modern LLMs.
3. The message the paper tries to convey is unclear, and some statements seem unjustified or out of the blue (see questions).
4. For an empirical paper, it is short of experiments and the empirical evidence is not convincing.

**Questions For Authors:**

1. How is “simple models which explain the training data generalize best" mentioned in the abstract related to the Occam's razor principle, and what does "explain the training data" mean?

2. In section 2.4 (which I believe is the main contribution of this paper) the summary part, you mentioined "... explicitly optimize a meta-learning objective that jointly minimizes training error and model complexity". However, the next-token prediction loss (or equivalently the prequential loss? Correct me if I'm wrong), is an upper bound for the training error plus model complexity, thus the statement doesn't seem valid?

3. In section 3.1 you mentioned two training objectives, one of them is to "training $T\_{\Phi}$ to predict past datapoints in the context". Why is this necessary? Can't a transformer model simply copy (or memorize) the datapoints in the context? It's

4. For the two different training objectives, do you implement different pretraining objectives or do you simply test the inference time ICL ability without pretraining?

5. In line 232 right column you stated "While nobody uses train-risk ICL in practice, it serves as an ideal control to illustrate our theory of ICL...". How does this train-risk illustrate your theory of ICL?

6. In figure2, why do you call the MSE error the "generalization error"?

7. I don't understand how could "simple" model be better at generalization in section 3.1 findings (althogh generalization itself is not formally defined in the paper). Is it simply due to the lack of expressivity of simple models so that overfitting is less likely?

8. In line 333 right column you mentioned "Not only is its prequential code length substantially higher than ...", are you suggesting that prequential code length is the same as the MSE loss since in figure 2c you used the generalization error (which is the MSE loss)?

9. In figure 2c, "our transformer" is pretrained on "a distribution of Mastermind tasks", could you provide some details of the training regime (i.e. the training data format, the training distribution etc.)?

**Relation To Broader Scientific Literature:**

The paper is related to unveiling the black box ICL of transformers.

**Theoretical Claims:**

There're no theoretical claims.

---

> ### Author Rebuttal · Authors · 2025-03-31
>
> Thank you for your constructive review.
>
> **Empirical studies**
>
> Our work is not an "empirical paper", but a theoretical one. Unlike standard theory papers, we studied 4 tasks including the challenging Mastermind task, multiple sequence models to highlight the generality of the theory (Fig. 2b), and conducted numerous ablations (Apx. D, E).
>
> Nevertheless, if this remains unconvincing, please clarify *which* conclusions (listed under the paper's “Findings.” headings) you find unsupported by our results or what additional experiments you’d liked to see.
>
> **Lack of theorems**
>
> We build on [1-3] to establish a novel connection between description length results for DNNs and ICL. In section 2, we mathematically derive the link from prequential code length (PCL) minimization to implicit model complexity when a pretrained model learns a task from examples presented in context. We chose not to use the "Theorem, Lemma, Proposition" formulation, as we believe it helps readability.
>
> **Relation between ICL and next-token prediction**
>
> ICL is indeed *not equivalent* to next-token prediction, and we do not suggest that it is. Rather, we show that training via next-token prediction loss explains the effectiveness of ICL because of the link from next-token prediction to compression.
>
> **Q1**
>
> Given equally good explanations, Occam's razor prefers the simplest one. In a typical ML textbook, Occam's razor is the justification for why simpler models generalize better (e.g., see chapter 1 in [4] and 7.3 in [10] for a detailed discussion), where "explain the training data" means minimizing training error. Further, the most common way to formalize model simplicity is using Kolmogorov complexity as we do. See [5-6] for short introductions, and [7] for a discussion in the context of deep learning.
>
> **Q2**
>
> As stated in L156-159, minimizing upper bounds to intractable quantities is standard practice in ML [4, chapter 10], e.g., we minimize the negative evidence lower-bound instead of the intractable negative log likelihood with VAEs [8] or diffusion models [9]. Furthermore, even when it is not being minimized through a meta-learning objective, PCL has been found to be a strong compression algorithm in deep learning settings [1], bounding $K(D, p)$ better than other methods.
>
> **Q3 and Q5**
>
> Our theory predicts that prequential ICL should generalize better since it was meta-learned to minimize model complexity as well as training error. To verify this, we need a ‘control’  that is meta-learned to only minimize training error, which is exactly "train-risk ICL". You're right that if implemented naively, such a train-risk learner could simply memorize in-context examples, which is why we minimally modify the Transformer architecture to summarize the context in a bottleneck. We explain this in L240-252 RHS.
>
> **Q4**
>
> We pretrain separate models using (a) the next-token prediction objective or (b) the past-token prediction objective on a distribution of tasks and then compare these resulting learners on unseen tasks at inference time (L252-258 and Apx. C.2). This is in line with other work studying ICL in controlled settings  (e.g., citations on L221-222 LHS)].
>
> **Q6**
>
> Generalization error (y-axis) at a particular context length (x-axis) is the inferred model's prediction error on *unseen* data (Fig. 2 caption). MSE is the standard metric used to measure error for regression.
>
> **Q7**
>
> Given equal training error, it is well-known that simple models generalize better [4, 10]. Overfitting indeed happens due to excess model complexity. We in fact do define generalization on L255-258 RHS as prediction error on unseen data, which is how it is always defined [4].
>
> **Q8**
>
> PCL and MSE are not the same. PCL is defined as the cumulative next-token negative log-likelihood (NLL) across a dataset (section 2.2,  Fig. 1, Eq. 4)—for regression, NLL is measured using MSE (well-known to be equivalent to NLL under a Gaussian). PCL is therefore given by the area under a curve in Fig. 2 (L300-301), whereas MSE is the y-axis of the plots for regression problems.
>
> **Q9**
>
> We describe the Mastermind task and data format in L237-247 LHS. Other training details are provided in Apx. C.
>
> [1] Blier & Ollivier (2018). The description length of deep learning models
>
> [2] Delétang et al. (2023). Language modeling is compression
>
> [3] Wilkenfeld (2019). Understanding as compression
>
> [4] Bishop & Nasrabadi (2006). Pattern recognition and machine learning
>
> [5] Nannen (2010). A short introduction to model selection, Kolmogorov complexity and Minimum Description Length
>
> [6] Wallace & Dowe (1999). Minimum message length and Kolmogorov complexity
>
> [7] Mingard et al. (2025). Deep neural networks have an inbuilt Occam's razor
>
> [8] Kingma & Welling (2013). Auto-encoding variational bayes
>
> [9] Song et al. (2020). Score-based generative modeling through stochastic differential equations
>
> [10] Shalev-Shwartz & Ben-David (2014). Understanding machine learning: From theory to algorithms

---

> > ### Comment · Reviewer_tMvj · 2025-04-04
> >
> > I thank the authors' for the responses.
> >
> > First, regarding Q2, while I acknowledge that "minimizing upper bounds to intractable quantities is standard practice in ML," this is not the case in the paper. The intractable quantities in question are the Kolmogorov complexities ( K(D|p)  and  K(p|T) ), while the proposed upper bound is the next-token prediction training loss, or equivalently PCL. Minimizing the training loss (the upper bound) does not necessarily equate to minimizing the Kolmogorov complexity (the intractable quantities), as the training objective is explicitly different from the Kolmogorov complexity. Therefore, the claim that "sequence models trained on cumulative next-token prediction losses explicitly optimize a meta-learning objective that jointly minimizes training error and model complexity" does not hold.
> >
> > Second, the main results in Sections 2.3 and 2.4 pertain to the training objective, which follows the standard LLM pretraining scheme, rather than ICL. Thus, I find the claim in the rebuttal that "training via next-token prediction loss explains the effectiveness of ICL" unconvincing. A proper connection between ICL and PCL would require an analysis during test time, as the core of ICL lies in the model’s ability to leverage context prompts at inference to generate high-quality responses for unseen inputs—despite being trained solely with next-token prediction.
> >
> > Given these concerns, I believe the paper requires substantial theoretical revisions, and I will maintain my current score.

---

> > > ### Author Response · Authors · 2025-04-05
> > >
> > > We thank the reviewer for their response.
> > >
> > > **Clarifying the usefulness of the bound**
> > >
> > > We apologize for the confusion here. We fully agree that we minimize PCL, which is an upper bound to the Kolmogorov complexity (K-complexity) and indeed if we reduce PCL by a certain quantity, it does not imply that we have reduced K-complexity by any quantity (it could even go up!).
> > >
> > > However, PCL ≥ K-complexity implies that if we have reduced PCL to some quantity $\epsilon$, we can with certainty say that K-complexity ≤ $\epsilon$. Therefore, if it is possible to minimize PCL through meta-learning (i.e., ICL), we can *guarantee* that the resulting model complexity + training error is small (at least as small as the PCL), whereas when minimizing training error (e.g., through standard maximum likelihood) we get no such guarantees (model complexity can be large and is never bounded).
> > >
> > > **Standard practice**
> > >
> > > We are confused by the reviewer’s following statement: “the training objective is explicitly different from the Kolmogorov complexity”. Even when the true objective (e.g., K-complexity) is explicitly different from the upper bound (e.g., PCL), it is still a valid strategy to minimize the bound in hope of minimizing the true objective, for example:
> > >
> > > $$ \log p(x; \theta) = \mathbb{E}\_{z \sim q\_\varphi(\cdot | x)}\left[\frac{\log p(x, z; \theta)}{q\_\varphi(z | x)}\right] + \mathbb{KL}\left[q\_\varphi(\cdot | x) || p(\cdot | x; \theta)\right] \geq \mathbb{E}\_{z \sim q\_\varphi(\cdot | x)}\left[\frac{\log p(x, z; \theta)}{q\_\varphi(z | x)}\right] $$
> > >
> > > Here, the right hand quantity (the Evidence Lower Bound) is also an explicitly different quantity to the left hand one (true objective); note that we switched the order because here the goal is to increase the true objective, not decrease. In particular, the true objective is not even a function of $\varphi$ but the right hand side is. Yet, training with this bound is a well-known practice in graphical models, especially VAEs, even though the bound is mostly not tight [1]. Even in this standard practice, one could be maximizing the ELBO in a way that does not necessarily maximize the log likelihood.
> > >
> > > Regarding K-complexity in particular, it is defined as the length of an optimally compressed string. Since it is about optimal compression, K-complexity is in fact *always* bounded using computable compression algorithms [2-4], such as PCL (a compression algorithm) as in our case.
> > >
> > > **Connection to ICL**
> > >
> > > The reviewer mentions that the “core of ICL lies in the model’s ability to leverage context prompts at inference to generate high-quality responses for unseen inputs” with which we completely agree. In our experiments, we are doing precisely this: the pre-trained model leverages context prompts (i.e., observations from a novel specification of the task) and generates high-quality responses (i.e., in line with the true underlying predictive model) for unseen inputs (unseen as the context came from a task unseen during training). Note that this analysis is during test time.
> > >
> > > The reviewer might be pointing to the fact that we study *example-based* ICL where the prompt contains demonstrations from a novel task, as opposed to the form of ICL in which the task instructions are linguistically described in a prompt given to an LLM. However, *both* example-based and instruction-based ICL are prevalently studied in the literature, and the term “ICL” does not refer to one in particular [5]—the important part is that contextual information in both cases describes novel tasks, and is used to learn at inference time. In fact, a number of works that aim to analyze and understand ICL follow similar example-based procedures [e.g., 6-8], and we make it clear early on in the introduction which form of ICL we study (lines 25-39 RHS).
> > >
> > > We thank the reviewer for their insight and would greatly appreciate an increase in rating if their concerns have been addressed.
> > >
> > > [1] Cremer, Chris, Xuechen Li, and David Duvenaud. "Inference suboptimality in variational autoencoders." *International conference on machine learning*. PMLR, 2018.
> > >
> > > [2] Nannen (2010). A short introduction to model selection, Kolmogorov complexity and Minimum Description Length
> > >
> > > [3] Blier & Ollivier (2018). The description length of deep learning models
> > >
> > > [4] Mingard et al. (2025). Deep neural networks have an inbuilt Occam's razor
> > >
> > > [5] Lampinen, A. K., Chan, S. C., Singh, A. K., & Shanahan, M. (2024). The broader spectrum of in-context learning
> > >
> > > [6] Zhang, Ruiqi, Spencer Frei, and Peter L. Bartlett. "Trained transformers learn linear models in-context." *Journal of Machine Learning Research* 25.49 (2024): 1-55.
> > >
> > > [7] Müller, Samuel, et al. "Transformers can do bayesian inference." *arXiv preprint arXiv:2112.10510* (2021).
> > >
> > > [8] Garg, Shivam, et al. "What can transformers learn in-context? a case study of simple function classes." *Advances in Neural Information Processing Systems* 35 (2022): 30583-30598.

---

### Official Review · Reviewer_grfc · 2025-03-17

**Overall Recommendation:** 3

**Summary:**

This paper explores a theoretical framework linking next-word prediction losses in large language models to coding-theoretic principles, often referred to as “prequenial” coding. It argues that simpler, more compact representations (inspired by Occam’s Razor) can facilitate stronger in-context learning performance. The authors present a bound-based objective aimed at improving generalization from prompts, and provide preliminary results on synthetic tasks suggesting the benefit of these simplicity-driven principles.

**Claims And Evidence:**

The submission posits that next-word prediction objectives parallel coding-based formulations and that enforcing an Occam’s Razor-style bound can yield practical improvements in in-context learning. The key claims are supported by derivations that connect model complexity and predictive accuracy, alongside synthetic experiments. However, the paper offers limited evidence of how these findings directly translate into real-world scenarios, since the experiments remain small-scale and somewhat specialized.

**Essential References Not Discussed:**

N/A

**Experimental Designs Or Analyses:**

The experiments use simple synthetic tasks designed to capture in-context learning behavior. They document improvements that align with the theoretical claims, but do not include extensive ablation or broader domain testing. Consequently, while the setup seems sound for initial validation, it provides limited evidence of robustness or real-world feasibility.

**Methods And Evaluation Criteria:**

The methods center on a bound-based reweighting scheme intended to emphasize simplicity in the learned representations. Evaluation is conducted via synthetic pattern extrapolation tasks, showing modest gains under the proposed approach. While these tasks illustrate potential effectiveness, more extensive benchmarks or realistic datasets would strengthen the case for broader applicability.

**Other Comments Or Suggestions:**

A broader discussion of how bound-based optimization might scale to real-world tasks and how it compares or integrates with standard in-context learning pipelines would considerably strengthen the paper. Small additions, like evaluating multiple domains or tasks, could showcase broader relevance.

**Other Strengths And Weaknesses:**

A notable strength is the novel theoretical perspective that attempts to unify coding-theoretic arguments with in-context learning. The main weaknesses are the lack of clarity regarding how these ideas translate into practical improvements, as well as the absence of a detailed analysis on the tightness of the proposed bounds.

**Questions For Authors:**

How do you envision scaling your bound-based approach to more complex, real-world tasks without incurring excessive computational cost? 2) Have you attempted to measure the tightness of your bounds empirically in different settings to ensure that they offer meaningful guidance rather than a loose theoretical construct? 3) Could you compare and contrast your approach with standard PAC-Bayes bounds to clarify any points of conceptual or methodological overlap?

**Relation To Broader Scientific Literature:**

By linking next-word prediction to coding-theoretic insights, this work resonates with the long-standing principle of minimum description length and various PAC-Bayes approaches, all of which emphasize model simplicity as a route to better generalization. It adds to recent discussions on in-context learning by proposing a formal perspective on why large language models can generalize from prompts.

**Theoretical Claims:**

The theoretical arguments rest on a newly introduced Occam’s Razor-inspired bound relating predictive cross-entropy to compact representations. The proofs, rooted in coding theory, appear consistent with standard generalization frameworks, but the paper does not delve into how tight or loose these bounds might be in practice, which raises questions about their practical utility.

---

> ### Author Rebuttal · Authors · 2025-03-31
>
> Thank you for your constructive review.
>
> **Clarification of contributions**
>
> The reviewer suggests that we “present a bound-based objective aimed at improving generalization from prompts” and that our work centres around “a newly introduced Occam’s Razor-inspired bound” that gives “stronger in-context learning performance”. We in fact do not attempt to improve ICL and do not propose novel methods for it: we provide a normative theory based on Occam’s razor to explain the effectiveness of ICL as it currently exists. Specifically, we argue that its effectiveness as a learning algorithm lies in the next-token prediction loss used to train sequence models, which optimizes a compression scheme called prequential coding (PCL) that implicitly fits simple models in-context that generalize well. We believe that this theoretical link is a novel contribution to the field of ICL.
>
> **Justification of experimental protocol**
>
> We decided to focus on synthetic tasks for a few reasons.
>
> 1. Interpretability:  For theoretical work, synthetic tasks are easier to control and results are easier to interpret, allowing us to concretely compare different objectives to illustrate the validity of our central insight: that ICL learners are more performant and efficient, especially in low-data regimes.
> 2. We are in a meta-learning setting where sequence models need to be trained on a large meta-distribution of tasks in order to perform ICL. With real-world data, it is difficult to control the size of the meta-distribution over tasks or find meta-distributions that are sufficiently broad.
> 3. Valid comparison against the train-risk ICL baseline for real LLM tasks we would have required training an LLM from scratch using the train-risk objective we outline (L223-238 RHS). Given that this paper is about a general theory of ICL, expending such industry-scale compute resources isn’t reasonable, especially considering that we perform ablations to carefully study modeling choices. Consequently, we experiment with non-iid HMM based tasks (L248-259 LHS) to capture the structure of natural language, following common practice (c.f., [1]).
> 4. Similar theoretical work in the field of ICL also makes use of synthetic tasks, and we aimed to remain consistent with standard practice (e.g., citations on L221-222 LHS).
>
> The reviewer suggests our work lacks “extensive ablation”, but we introduce baselines (e.g., train-risk ICL, off-the-shelf learners with different inductive biases) to isolate the role of the next-token loss that bounds $K(D,p)$.
>
> The reviewer suggests a lack of “multiple domains or tasks”, but our experimental settings are in fact wide-ranging: we study linear & nonlinear tasks, regression & classification, iid & non-iid, scaled task difficulties, Transformers & SSMs, and models trained from scratch & LLMs.
>
> **Q1**
>
> We would again like to emphasize that we did not introduce a novel training algorithm for sequence models as the reviewer suggests. We provided a theory that explains why the standard next-token prediction loss is an effective method for training sequence models. We do not have to “scale our approach to more complex, real-world tasks” since cumulative next-token prediction loss is *already* the objective used to train LLMs.
>
> **Q2**
>
> Measuring the tightness of the bound $K(D,p) \lt L_{preq}$  involves computing Kolmogorov complexity, which is uncomputable. However, minimizing an upper bound on an intractable quantity follows a long line of work in ML. For example, as we state in our paper (L156-159 RHS), all variational inference methods that minimize the negative ELBO—from VAEs to diffusion models—learn via a tractable bound to an intractable quantity (the negative log likelihood). In compression too, there is longstanding work (c.f., [2]) proposing variational approximations to minimizing the complexity of deep learning models. Pushing down an upper-bound is a workable proxy for optimizing a target quantity, therefore our argument that minimizing PCL minimizes training error + model complexity is valid. Finally, even when it is not being minimized through a meta-learning objective, PCL has been found to be a strong compression algorithm in deep learning settings [3], bounding $K(D, p)$ better than other methods. We will further clarify this in revisions.
>
> **Q3**
>
> PAC-Bayes bounds are also rooted in Kolmogorov complexity, but require a prior over models. PCL only depends on a learning algorithm. Viewing a sequence model as a learning algorithm therefore makes it easy to compute PCL, but not PAC-Bayes bounds. We’ll include a brief discussion about the link in the revised draft.
>
> [1] Xie, S. M., Raghunathan, A., Liang, P., & Ma, T. (2021). An explanation of in-context learning as implicit bayesian inference
>
> [2] Honkela, A., & Valpola, H. (2004). Variational learning and bits-back coding: an information-theoretic view to Bayesian learning
>
> [3] Blier, L., & Ollivier, Y. (2018). The description length of deep learning models

---

### Official Review · Reviewer_dU6f · 2025-03-22

**Overall Recommendation:** 3

**Summary:**

The paper studies the problem of incontext learning(ICL) and establishes a connection between ICL and Occam's Razor, arguing that next token prediction used in training Transformers implicitly minimizes both the training error and complexity of the learned model. The authors show that this joint objective is equivalent to prequential coding and the meta-learning problem of minimizing the prequential code length is already solved by next token prediction objective used in training ICL. Through both theoretical analysis and experimental results, the authors argue that ICL implicitly favors simpler model that generalize well which is in line with the principle of Occam's Razor.

**Claims And Evidence:**

The claims made in the paper appear to be supported by convincing theoretical and empirical evidence though I have not verified the theoretical results.

**Essential References Not Discussed:**

NA

**Experimental Designs Or Analyses:**

The authors conducted experiments to compare ICL to standard trianing error minimization using SGD. They also conducted the impact of transformer architecture on prequential code minimization and their generalization ability which is highly appreciative.

**Methods And Evaluation Criteria:**

The authors perform experiments using synthetic tasks that allow finer control over task and sample complextiy. The evaluation criteria of prequential length and generalization error are very relevant to the problem at hand.

**Other Comments Or Suggestions:**

It would be valuable to include some more realistic tasks in empirical evaluation.

**Other Strengths And Weaknesses:**

Strengths: The paper establishes a good theoretical connection between the joint objective of reducing training error and model compression to prequential encoding and establishes the meta-learning problem of next token prediction used to train ICL already solves this problem.

Weakness: The connection between prequential encoding and ICL meta-training is more of an upper bound rather than a direct equivalence. The prequential encoding does not directly suggest the specific meta-training algorithm or architectures used.

**Questions For Authors:**

1. You highlight that current ICL methods underfit. Based on your study, do you see a way to mitigate this issue perhaps by better architecture or model design?
2. Have you conducted experiments on real world task involving natural languages? If so, could you please elaborate on that?

**Relation To Broader Scientific Literature:**

ICL has been widely used in modern LLMs to solve various task without model fine tuning and is an important paradigm of study in modern LLMs. This paper takes an important step in properly studying ICL from the lens of model compression and Occam's razor.

**Theoretical Claims:**

I have not verified the theoretical correctness of proofs.

---

> ### Author Rebuttal · Authors · 2025-03-31
>
> Thank you for your constructive review.
>
> **Minimizing a tractable upper bound on an intractable objective is valid and common in ML**
>
> > The connection between prequential encoding and ICL meta-training is more of an upper bound rather than a direct equivalence.
> >
>
> The reviewer is correct in pointing out the upper bound given by prequential code length (PCL). It would be ideal to directly minimize Kolmogorov complexity, though this is widely known as being intractable.
>
> In contrast, we minimize an upper bound (with learnable parameters $T_\phi$), following a long line of work in ML. For example, as we state in our paper (L156-159 RHS), all variational inference methods that minimize the negative evidence lower bound (ELBO)—from variational auto-encoders to diffusion models—learn via a tractable bound to an intractable quantity (the negative log likelihood). In compression too, there is longstanding work (c.f., [1]) proposing variational approximations to minimizing the complexity of deep learning models. Pushing down the upper-bound is a workable proxy for optimizing a target quantity, therefore our argument that minimizing PCL minimizes training error + model complexity is valid. Finally, even when it is not being minimized through a meta-learning objective, PCL has been found to be a strong compression algorithm in deep learning settings [2], bounding $K(D, p)$ better than other methods. We will further clarify this important point in revisions.
>
> **Prequential coding through next-token prediction with sequence models abstracts over meta-training algorithm and meta-learner architecture**
>
> > The prequential encoding does not directly suggest the specific meta-training algorithm or architectures used.
> >
>
> It is an advantage of our theory to provide a compression-based account of ICL for sequence models without being specific to particular architecture choices: indeed, we show that the theory holds for both Transformers and state-space-models (Section 3.3). Our message is that having a PCL minimization objective—regardless of architecture or optimization details—is good practice. Further, our findings identify two significant challenges, which we highlighted in our experiments section:
>
> 1. Underfitting by in-context models as a result of limited sequence model expressivity and compute power compared to training DNNs from scratch on a task (sections 3.2, 3.3)
> 2. Meta-generalization of the sequence model to novel tasks (section 3.4)
>
> In section 5, we did in fact discuss possible approaches for addressing these challenges, some of which have been explored in prior work (L385-431 RHS) and some of which are novel ideas. We believe that solving these challenges is out of scope for our current theoretical work, but that our theory is useful for both understanding and addressing them. We will further clarify these points in the discussion.
>
> **Q1**
>
> Based on our theory and results, we outlined a promising approach to the ICL underfitting problem in L385-413 RHS.
>
> **Q2**
>
> We decided to focus on synthetic tasks for a few reasons.
>
> 1. Interpretability:  For theoretical work, synthetic tasks are easier to control and results are easier to interpret, allowing us to concretely compare different objectives to illustrate the validity of our central insight: that ICL learners are more performant and efficient, especially in low-data regimes.
> 2. We are in a meta-learning setting where sequence models need to be trained on a large meta-distribution of tasks in order to perform ICL. With real-world data, it is difficult to control the size of the meta-distribution over tasks or find meta-distributions that are sufficiently broad.
> 3. For a valid comparison against the train-risk ICL baseline, for real LLM tasks, we would have needed to train an LLM from scratch using the train-risk objective we outline (L223-238 RHS). Given that this paper is about a general theory about ICL, expending such industry-scale compute resources isn’t reasonable, especially considering that we perform ablations to carefully study modeling choices. Consequently, we experiment with non-iid HMM based tasks (L248-259 LHS) to capture the structure of natural language, following common practice (c.f., [3]).
> 4. Similar theoretical work in the field of ICL also makes use of synthetic tasks, and we aimed to remain consistent with standard practice (e.g., citations on L221-222 LHS).
>
> [1] Honkela, A., & Valpola, H. (2004). Variational learning and bits-back coding: an information-theoretic view to Bayesian learning
>
> [2] Blier, L., & Ollivier, Y. (2018). The description length of deep learning models
>
> [3] Xie, S. M., Raghunathan, A., Liang, P., & Ma, T. (2021). An explanation of in-context learning as implicit bayesian inference

---

### Decision · Program_Chairs · 2025-05-01

**Decision:**

Accept (poster)

**Comment:**

This submission explores a theoretical perspective linking in-context learning (ICL) with Occam’s razor through the lens of prequential coding. Most reviewers found the paper’s central insight, that the cumulative next-token prediction objective implicitly encourages simpler learned models that exhibit strong generalization in context, interesting and grounded in a sound line of literature on description length and Kolmogorov complexity. The authors supported their argument with controlled synthetic experiments that illustrate the difference between meta-training objectives that account for model complexity (prequential) and those that only minimize training error. Two reviewers gave “weak accept” recommendations, citing the paper’s coherent theoretical message and its plausible, though modestly scaled, empirical support. They also saw value in the discussion of underfitting and the paper’s potential to spur new research into improved sequence-model architectures for in-context learning.

However, one reviewer recommended rejection, contending that the theory lacks the typical “theorem-proof” format and that linking pretraining-time next-token prediction strictly to in-context generalization remains incomplete without a tighter test-time analysis or more direct empirical verification at scale. This reviewer further argued that minimizing a practical upper bound (i.e., prequential code length) does not necessarily imply minimal Kolmogorov complexity, so the Occam’s razor claim may be overstated. In their rebuttal, the authors clarified that bounding Kolmogorov complexity with a tractable surrogate remains standard practice in machine learning (e.g., variational methods) and emphasized that in-context generalization is indeed assessed through example-based prompts at inference. While concerns persist regarding the gap between real-world LLM usage and the paper’s synthetic tasks, the consensus is that this work offers an insightful theoretical perspective.